# Improving Compositional Generation with Diffusion Models Using Lift Scores

**Chenning Yu** [1]   **Sicun Gao** [1]

## Abstract

We introduce a novel resampling criterion using lift scores, for improving compositional generation in diffusion models. By leveraging the lift scores, we evaluate whether generated samples align with each single condition and then compose the results to determine whether the composed prompt is satisfied. Our key insight is that lift scores can be efficiently approximated using only the original diffusion model, requiring no additional training or external modules. We develop an optimized variant that achieves relatively lower computational overhead during inference while maintaining effectiveness. Through extensive experiments, we demonstrate that lift scores significantly improved the condition alignment for compositional generation across 2D synthetic data, CLEVR position tasks, and text-to-image synthesis. Our code is available at rainorangelemon.github.io/complift.

## 1. Introduction

Despite the success of diffusion models in generating high-quality images (Sohl-Dickstein et al., 2015; Ho et al., 2020; Nichol & Dhariwal, 2021; Song et al., 2020), they sometimes fail to produce coherent and consistent results when the condition is complex or involves multiple objects (Huang et al., 2023; Feng et al., 2022; Chefer et al., 2023). This limitation is particularly evident in tasks that generate the images with specific combinations of attributes or objects, where the generated samples may only partially satisfy the conditions or exhibit inconsistencies across different attributes.

To mitigate this issue, several methods have been proposed to improve the compositional generation with diffusion models. These methods can be described as 2 main categories: (1) **training-based** methods, which train or finetune the diffusion model to improve its ability to generate samples that satisfy complex prompts (Huang et al., 2023; Bao et al., 2024), and (2) **training-free** methods, which either refine the sampling process, or apply rejection sampling to refine the generated samples without modifying the underlying model (Feng et al., 2022; Chefer et al., 2023). These 2 categories of methods are not necessarily mutually exclusive, and they can be combined to achieve better performance.

We introduce *CompLift*, a training-free rejection criterion for improving compositional generation in diffusion models. The concept of lift score (Brin et al., 1997) builds on a simple observation: if a generated sample truly satisfies a condition, the model should perform better at denoising it when given that condition compared to unconditional denoising. This insight leads to an efficient criterion for accepting or rejecting samples. Similar to Diffusion Classifier (Li et al., 2023), we show that lift scores can be efficiently approximated using only the original diffusion model's predictions, requiring no additional training or external modules. By decomposing the satisfaction of a complex prompt into the satisfaction of multiple simpler sub-conditions using lift scores, our approach can boost the model's alignment with complex prompts without changing the sampling process. Through extensive experiments on 2D synthetic data, CLEVR position tasks, and text-to-image generation, we demonstrate that *CompLift* significantly improves compositional generation while maintaining computational efficiency.

**Contributions.** Our contributions are threefold: ❶ We introduce *CompLift*, as a novel resampling criterion using lift scores for compositional generation, requiring no additional training (Section 3). ❷ We explore the design space of *CompLift*, including noise sampling strategies, timestep selection, and caching techniques to optimize computational efficiency (Section 4). ❸ We scale our approach to text-to-image generation, demonstrating its effectiveness in improving compositional generation in this challenging domain (Sections 5 and 6).

## 2. Related Work

Recent studies have revealed significant limitations in diffusion models (Dhariwal & Nichol, 2021) for handling compositional generation, particularly as missing objects, incorrect spatial relationships, and attribute inconsistencies (Huang

---

[1]Department of Computer Science and Engineering, UC San Diego, La Jolla, USA. Correspondence to: Chenning Yu <chy010@ucsd.edu>, Sicun Gao <sicung@ucsd.edu>.

*Proceedings of the 42nd International Conference on Machine Learning*, Vancouver, Canada. PMLR 267, 2025. Copyright 2025 by the author(s).

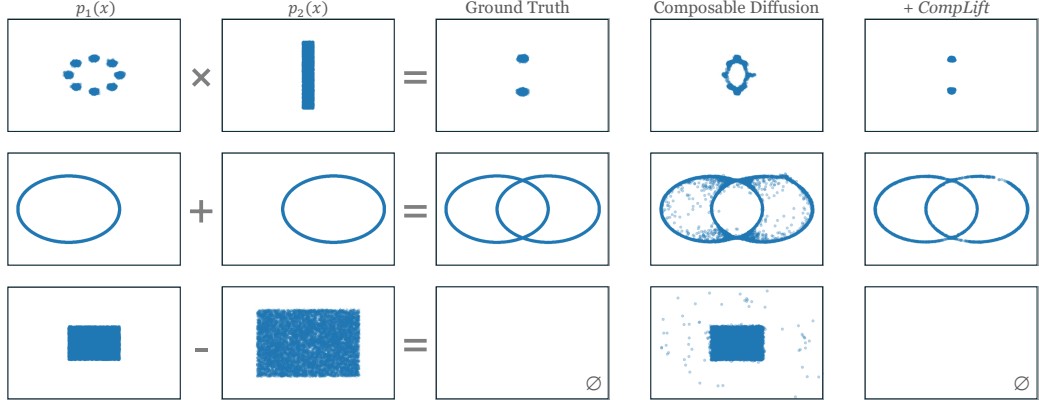

*Figure 1.* **An illustration of product, mixture, and negation compositional models, and the improved sampling performance using *CompLift*.** Left to right: Component distributions, ground truth composed distribution, composable diffusion samples, samples accepted by *CompLift*. Top: product, center: mixture, bottom: negation. ∅ represents the empty set - no samples are generated or accepted. Each component distribution is trained independently using a 2D score-based diffusion model. Accuracy is evaluated based on whether generated samples fall into the support or within the $3\sigma$ region of the composed distribution (details in Section 6, Appendix E).

et al., 2023; Feng et al., 2022; Chefer et al., 2023).

Current approaches to address these challenges fall into several categories. One line of work leverages the *cross-attention* mechanisms, modifying attention values during synthesis to emphasize overlooked tokens (Feng et al., 2022; Chefer et al., 2023; Wu et al., 2023; Wang et al., 2024). Another approach introduces *layout control* for precise spatial arrangement of objects (Feng et al., 2024; Chen et al., 2024). While effective, these methods require specific model architectural knowledge, unlike our model-agnostic approach.

The *diffusion process* itself can also be modified to improve compositional generation. Composable Diffusion (Liu et al., 2022) guides generation by combining individual condition scores. Discriminator Guidance (Kim et al., 2022) employs an additional discriminator to reduce the distribution mismatch of the generated samples. Recent methods introduce advanced sampling techniques like Markov Chain Monte-Carlo (Du et al., 2023), Restart Sampling (Xu et al., 2023), rejection sampling (Na et al., 2024), and particle filtering (Liu et al., 2024). We complement these methods by providing a technique that can be integrated without assumptions about the underlying sampling process.

Most closely related to our work are *resampling strategies* that operate as either selection of the final image (Karthik et al., 2023) or search of the initial noise (Qi et al., 2024; Ma et al., 2025). However, these methods typically require external models for quality assessment. In contrast, our approach leverages the denoising properties of the diffusion model itself (Li et al., 2023; Chen et al., 2023; Kong et al., 2023), eliminating the need for additional models.

Our work is not the first to apply lift scores to diffusion models. Condition Alignment Score (Hong et al., 2024) proposes an equivalence to rank the alignment of images with a whole single prompt. Kong et. al (Kong et al., 2023) regards lift scores as point-wise mutual information for the explanability of the generated images. In this work, we investigate the use of lift scores as *compositional criterion* for rejection and resampling in compositional generation.

## 3. Using Lift Scores for Rejection Sampling

### 3.1. Diffusion Model Preliminaries

Diffusion model is a class of generative models that generate samples by denoising a noisy input iteratively. Given a clean sample $\boldsymbol{x}_0$, the forward diffusion process $q(\boldsymbol{x}_t|\boldsymbol{x}_{t-1})$ sequentially adds Gaussian noise to $\boldsymbol{x}_{t-1}$ at each timestep $t$, whereas the learned reverse diffusion process $p_\theta(\boldsymbol{x}_{t-1}|\boldsymbol{x}_t, \boldsymbol{c})$ denoises $\boldsymbol{x}_t$ to reconstruct $\boldsymbol{x}_{t-1}$, optionally conditioned on a condition $\boldsymbol{c}$. The conditional probability of $\boldsymbol{x}_0$ can be defined as:

$$p_\theta(\boldsymbol{x}_0|\boldsymbol{c}) = \int_{\boldsymbol{x}_{1:T}} p(\boldsymbol{x}_T) \prod_{t=1}^{T} p_\theta(\boldsymbol{x}_{t-1}|\boldsymbol{x}_t, \boldsymbol{c}) \, d\boldsymbol{x}_{1:T},$$

where $p(\boldsymbol{x}_T)$ is typically a standard Gaussian $\mathcal{N}(\boldsymbol{0}, \mathbf{I})$.

The diffusion model is often parameterized as a neural network $\epsilon_\theta$ to represent the denoising function. Using the fact that $q(\boldsymbol{x}_t|\boldsymbol{x}_0) := \mathcal{N}(\boldsymbol{x}_t; \sqrt{\bar{\alpha}_t}\boldsymbol{x}_0, (1 - \bar{\alpha}_t)\mathbf{I})$, the model is trained by maximizing the variational lower bound (VLB) of the log-likelihood of the data, which can be approximated as the Evidence Lower Bound (ELBO), i.e., diffusion loss:

$$\log p_\theta(\boldsymbol{x}_0|\boldsymbol{c}) \geq \mathbb{E}_q \left[ \log \frac{p_\theta(\boldsymbol{x}_{0:T}, \boldsymbol{c})}{q(\boldsymbol{x}_{1:T}|\boldsymbol{x}_0)} \right]$$

$$= -\mathbb{E}_{\epsilon, t} \left[ w_t \|\epsilon - \epsilon_\theta(\boldsymbol{x}_t, \boldsymbol{c})\|^2 \right] + C.$$

Following the previous works (Ho et al., 2020; Li et al., 2023; Chen et al., 2023), we set $w_t = 1$ for all $t$ as it achieves good performance in practice.

**Algorithm 1** Rejection with *CompLift*

1: **Input:** test sample $x_0$, condition $\{c_i\}_{i=1}^n$, # of trials $T$, algebra $\mathcal{A}$
2: Initialize $\texttt{Lift}[c_i] = \text{list}()$ for each $c_i$
3: **for** trial $j = 1, \ldots, T$ **do**
4:    Sample $t \sim [1, 1000]$; $\epsilon \sim \mathcal{N}(0, I)$
5:    $x_t = \sqrt{\bar{\alpha}_t} x_0 + \sqrt{1 - \bar{\alpha}_t} \epsilon$
6:    **for** condition $c_k \in \{c_i\}_{i=1}^n$ **do**
7:       $lift_j(x_0|c_i) = \|\epsilon - \epsilon_\theta(x_t, \varnothing)\|^2 - \|\epsilon - \epsilon_\theta(x_t, c_k)\|^2$
8:       $\texttt{Lift}[c_k].\text{append}(lift_j(x_0|c_i))$
9:    **end for**
10: **end for**
11: $\{lift(x_0|c_i)\}_{i=1}^n = \{\texttt{mean}(\texttt{Lift}[c_i])\}_{i=1}^n$
12: **return** $\texttt{Compose}(\{lift(x_0|c_i)\}_{i=1}^n, \mathcal{A})$    ▷ Table 1

| Type | Algebra | Acceptance Criterion |
|------|---------|----------------------|
| Product | $c_1 \wedge c_2$ | $\min_{i \in [1,2]} lift(x|c_i) > 0$ |
| Mixture | $c_1 \vee c_2$ | $\max_{i \in [1,2]} lift(x|c_i) > 0$ |
| Negation | $\neg c_1$ | $lift(x|c_1) \leq 0$ |

*Table 1.* **Examples of** $\texttt{Compose}$ **for multiple conditions.**

## 3.2. Approximating Lift Scores

Given a sample $x$, we wish to check whether it aligns with a condition $c$. In other words, we want to see how well the association $x$ has with $c$. This brings us to the concept of *lift*, which is an existing concept in data mining:

$$lift(x|c) := \log \frac{p(c|x)}{p(c)} = \log \frac{p(x,c)}{p(x)p(c)} = \log \frac{p(x|c)}{p(x)}.$$

We note that the definition of lift scores is slightly different from the traditional data mining definition (Brin et al., 1997) as it is defined in the logarithmic space. We abuse the notation since it is easier to implement in the logrithm space, and we call it *lift score* throughout this paper.

One perspective to understand lift scores is to regard $p$ as a perfectly-learned classifier. If $x$ is an arbitrary unknown sample, this classifier tries its best to predict the probability of $x$ belonging to class $c$ as $p(c)$. However, once $x$ is given as a known sample, the classifier will utilize the information in $x$ to increase its prediction accuracy, as $p(c|x)$. If $x$ has any positive association with $c$, we would expect $p(c|x) > p(c)$, which means $lift(x|c) > 0$.

**Approximating the lift scores.** In preliminaries, we see that both $p(x|c)$ and $p(x)$ can be approximated using ELBO:

$$p(x|c) \approx \exp(-\mathbb{E}_{t,\epsilon}\|\epsilon - \epsilon_\theta(x_t, c)\|^2 + C), \quad (1)$$

$$p(x) \approx \exp(-\mathbb{E}_{t,\epsilon}\|\epsilon - \epsilon_\theta(x_t, \varnothing)\|^2 + C). \quad (2)$$

Notice that the constant $C$ is independent of $x$ and $c$, therefore dividing $p(x|c)$ by $p(x)$ cancels $C$. This gives us the approximation of *lift* in the form of $\log \frac{p(x|c)}{p(x)}$:

$$lift(x|c) \approx \mathbb{E}_{t,\epsilon}\{\|\epsilon - \epsilon_\theta(x_t, \varnothing)\|^2 - \|\epsilon - \epsilon_\theta(x_t, c)\|^2\}. \quad (3)$$

Equation (3) gives us another explanation of lift scores from a denoising perspective: if $c$ is positively associated with $x$, then given a sample $x_t$ by adding noise to $x$, a diffusion

model should improve the reconstruction quality to $x$ from the noisy $x_t$ if $c$ is given to the model, compared to the reconstruction quality of $x_t$ when $c$ is not known. Hence, **the criterion of *CompLift* for single condition is:**

$$\pi(x|c) = \begin{cases} 1 \text{ (accept)} & \text{if Equation (3)} > 0; \\ 0 \text{ (reject \& resample)} & \text{if Equation (3)} \leq 0. \end{cases}$$

The $\texttt{Compose}$ function encodes the logical structure of the prompt. This compositional logic mirrors logical inference over sub-condition satisfaction and enables generalization to complex prompts (see examples in Figure 1 and Table 1). For instance:
**(product)** $\texttt{Compose}$ accepts the sample only if all component conditions have positive lift scores.
**(mixture)** The sample is accepted if at least one lift score is positive.
**(negation)** The sample is rejected if the corresponding lift score is positive.

We describe the general form of $\texttt{Compose}$ in Appendix B.

## 4. Design Space of *CompLift*

We present the naive version of *CompLift* in Algorithm 1, where the lift score in Equation (3) is estimated using Monte-Carlo sampling. In this section, we discuss design choices of *CompLift*. We wish to answer the following questions:
**(Section 4.1)** How does the noise sampling affect the estimation accuracy of lift scores?
**(Section 4.2)** What role do different timesteps play in the effectiveness of our criterion?
**(Section 4.3)** Can we optimize the computation overhead via caching strategies?

**A Running Example: 2D Synthetic Dataset.** To illustrate the behavior of *lift scores* and the effect of different design choices, we use synthetic 2D datasets composed from component distributions such as Gaussian mixtures on a ring or uniform strips. Each composed distribution is defined by applying algebraic combinations to these components (see Figure 1 as an example). Samples are evaluated based on accuracy: whether they fall within the support of the target composed distribution. We refer readers to Section 6 and Appendix E for training details and metrics.

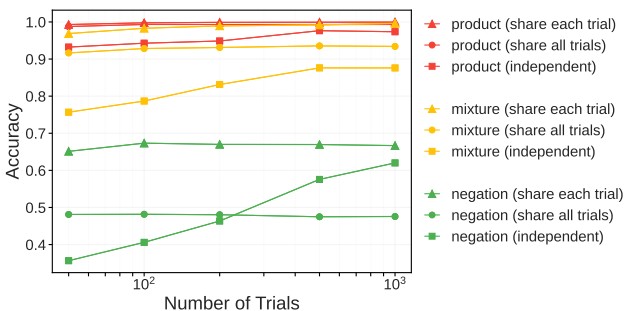

*Figure 2.* **The accuracy of *CompLift* with different noise sampling strategies on 2D synthetic dataset.** See Section 4.1.

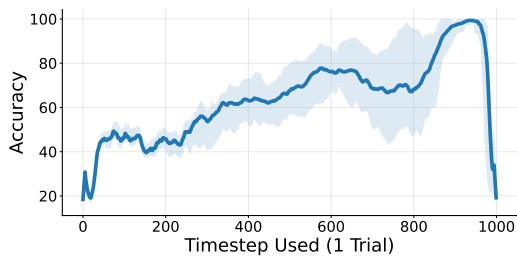

*Figure 3.* **Accuracy of acceptance/rejection over a single sampled timestep for pretrained model (Liu et al., 2022) on CLEVR Position dataset.** We found that models trained with importance sampling require importance sampling for ELBO estimation.

## 4.1. Effect of Noise

In Line 7 in Algorithm 1, we choose to share the $\epsilon$ when estimating both $p(\boldsymbol{x}|\boldsymbol{c})$ and $p(\boldsymbol{x})$. To validate this design, we conduct the experiment on a 2D synthetic dataset with the following 3 settings:

(1) **Independent Noise**: We sample two independent noises $\epsilon$ and $\epsilon'$ for $p(\boldsymbol{x}|\boldsymbol{c})$ and $p(\boldsymbol{x})$, respectively.
(2) **Shared Noise for Each Trial**: We share the noise $\epsilon$ between $p(\boldsymbol{x}|\boldsymbol{c})$ and $p(\boldsymbol{x})$, each trial samples a new noise.
(3) **Shared Noise Across All Trials**: We share the noise $\epsilon$ across all trials for both $p(\boldsymbol{x}|\boldsymbol{c})$ and $p(\boldsymbol{x})$.

In Figure 2, we observe that sharing noise for each trial is mostly robust over the number of trials, which is consistent with the conclusion in Diffusion Classifier (Li et al., 2023). We choose to use this design as the default setting. The independent strategy increases the accuracy with more trials, especially for mixture and negation algebra. We also find a small regression in the sharing strategies across all tasks. Our hypothesis is that sharing the same noise introduces some bias in the estimation, and it may over-reject samples as a conservative way. With more trials, the bias is amplified, thus makes more samples over rejected.

## 4.2. Effect of Timesteps

Similar to Diffusion Classifier (Li et al., 2023), we found that in most cases, if diffusion model $\epsilon_\theta$ is trained with

---

**Algorithm 2** Optimized Rejection with *Cached CompLift*

1: **Input:** conditions $\{\boldsymbol{c}_i\}_{i=1}^n$, algebra $\mathcal{A}$, # of reverse timesteps $N$
2: Initialize $\boldsymbol{x}_T \sim \mathcal{N}(0, I)$
3: **for** timestep $t_j \in [t_N, t_{N-1}, \ldots, t_1]$ **do**
4:     ▷ Standard generation process
5:     $\boldsymbol{x}_{t_{j-1}} = \texttt{step}(\boldsymbol{x}_{t_j}, \epsilon_\theta, \{\boldsymbol{c}_i\}_{i=1}^n)$
6:     ▷ Add predictions to cache
7:     **if** classifier-free guidance **then**
8:        Add precomputed $\epsilon_\theta(\boldsymbol{x}_{t_j}, \varnothing)$ to cache
9:     **end if**
10:    **if** Composable Diffusion **then**
11:       Add precomputed $\{\epsilon_\theta(\boldsymbol{x}_{t_j}, \boldsymbol{c}_i)\}_{i=1}^n$ to cache
12:    **end if**
13:    Add $\boldsymbol{x}_{t_j}$ to cache
14: **end for**
15: **run** Algorithm 1 with cache, where # of trials $T = N$, and $\boldsymbol{x}_{t_j}$, $\epsilon_\theta(\boldsymbol{x}_{t_j}, \varnothing)$, and $\epsilon_\theta(\boldsymbol{x}_{t_j}, \boldsymbol{c}_i)$ are reused.

---

uniform timestep sampling, sampling the timestep uniformly for ELBO estimation (Line 4 in Algorithm 1) is sufficient.

More surprisingly, we also found that the timestep for ELBO estimation needs importance sampling when the diffusion model is trained under importance sampling, where the importance sampling enhances the stability of optimizing $L_{\text{VLB}}$ (Nichol & Dhariwal, 2021). In particular, we discovered a pretrained model for CLEVR Position dataset (Liu et al., 2022) requires the importance sampling because of the above reason. We show the effect in Figure 3.

This suggests that the timestep sampling of ELBO estimation to stay consistent with the sampling strategy at training stage. Therefore, we choose to use importance sampling for the CLEVR pretrained model, and use uniform sampling as default for other tasks, i.e., 2D dataset and text-to-image.

## 4.3. Improving Computational Efficiency with Cached Prediction

In the naive version of *CompLift* in Algorithm 1, the algorithm requires $(n + 1) \cdot T$ times of forward passes for $n$ conditions. To reduce the computational cost, we propose caching the diffusion model's intermediate predictions.

We find that we can reuse the prediction of $\epsilon_\theta(\boldsymbol{x}_t, \varnothing)$ and $\{\epsilon_\theta(\boldsymbol{x}_t, \boldsymbol{c}_i)\}_{i=1}^n$ at intermediate timestep $t$ during the generation process of $\boldsymbol{x}_0$. Note that the unconditional prediction $\epsilon_\theta(\boldsymbol{x}_t, \varnothing)$ is typically precomputed for generation with classifier-free guidance (CFG) (Ho & Salimans, 2022). For conditional prediction $\{\epsilon_\theta(\boldsymbol{x}_t, \boldsymbol{c}_i)\}_{i=1}^n$, they are precomputed under the Composable Diffusion framework (Liu et al., 2022). Under this framework, we can reduce the number of extra forward passes from $(n + 1) \cdot T$ to 0.

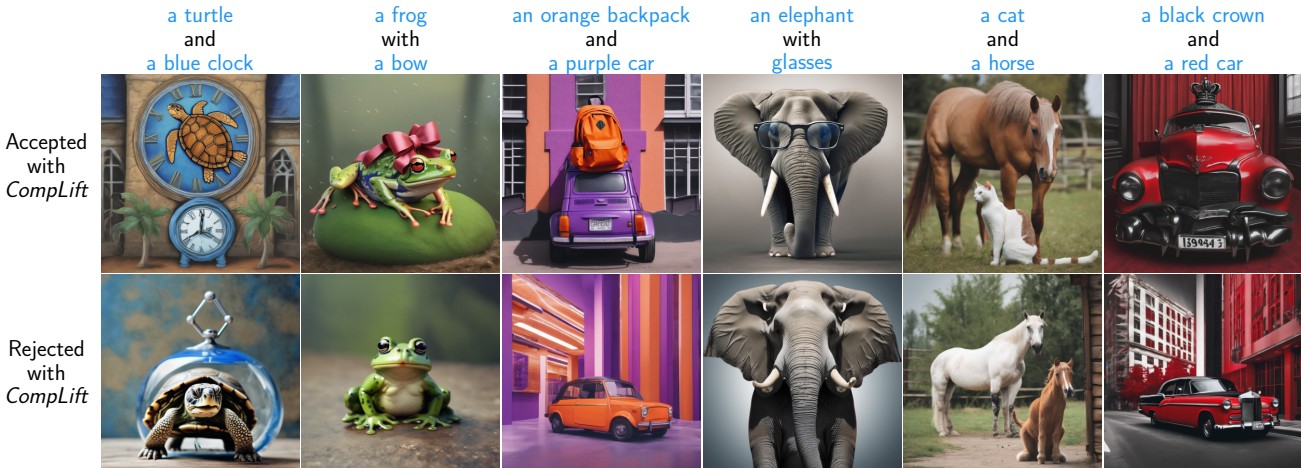

*Figure 4.* **Accepted and rejected SDXL examples using *CompLift* criterion.** Objects in blue are composed through the given prompts. Here we show the images from the text prompts with the most-improved CLIP scores. See more examples in Appendix G.

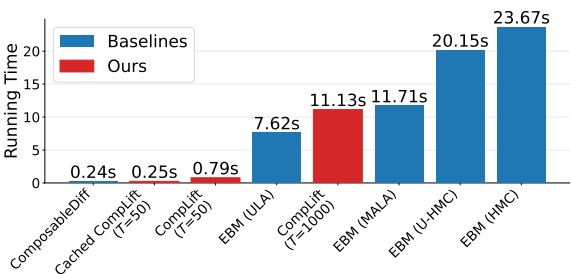

*Figure 5.* **Average running time on 2D synthetic dataset.** $T$ indicates number of trials. Note that our methods can be further optimized to the latency of only 1 forward pass with parallelization, while MCMC methods require sequential computation. See Table 2 for performance comparison of all the methods.

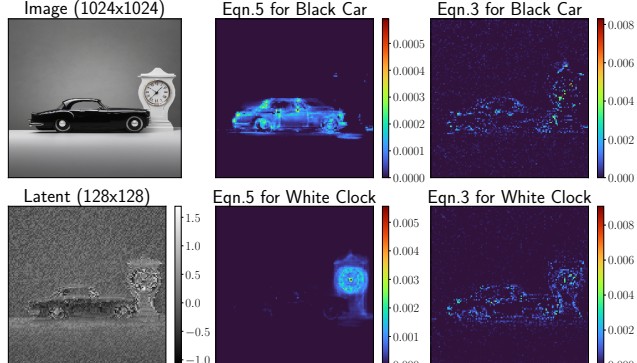

*Figure 6.* **Replacing $\epsilon$ in the differential loss of Equation (3) by $\epsilon_\theta(\boldsymbol{x}, \boldsymbol{c}_{\text{compose}})$ reduces the estimation variance.** Prompt: *a black car and a white clock*. Pixels with negative scores are masked out.

For more general framework, computing the conditional predictions $\{\epsilon_\theta(\boldsymbol{x}_t, \boldsymbol{c}_i)\}_{i=1}^n$ as additional computation on-the-fly during generation is beneficial for caching. With a sufficient amount of GPU memory, this operation typically would not bring extra time overhead, since all computations can be done in parallel to the original generation task. In this context, we can reduce the number of extra forward passes to $n \cdot T$, if CFG is used.

We provide the optimized version of *CompLift* in Algorithm 2. When running Line 4 in Algorithm 1, $\epsilon$ is not i.i.d. any more, but computed as $\epsilon = (\boldsymbol{x}_t - \sqrt{\bar{\alpha}_t}\boldsymbol{x}_0)/\sqrt{1 - \bar{\alpha}_t}$ instead using the cached $\boldsymbol{x}_t$. Another difference is that $T$ is constrained to be exactly the same as a number of inference steps $N$. In practice, we found that these 2 changes do not lead to a significant performance drop as long as $T$ is not too small. We show the comparison of the running time on 2D synthetic task in Figure 5. The overhead introduced by the cached *CompLift* is negligible for the Composable Diffusion baseline (Liu et al., 2022). With enough GPU memory, our method can be further optimized to the latency

of only 1 forward pass by parallelization, while MCMC methods (Du et al., 2023) require sequential computation. For more details, see pseudo-code in Appendix D.

## 5. Scaling to Text-to-Image Generation

We mainly consider the problem of **missing objects** in the text-to-image compositional task. Given multiple objects $\{\boldsymbol{c}_i\}_{i=1}^n$ to address simultaneously, a prominent error of diffusion models is that the generated images sometimes do not include object(s) $\{\boldsymbol{c}_j | j \in [1, n]\}$ mentioned in the text.

### 5.1. Counting Activated Pixels as Existence of Objects

Our approach is to count the number of activated pixels in the latent space to determine the existence of objects. A pixel at position $[a, b]$ in the latent $\boldsymbol{z}$ is activated for condition $\boldsymbol{c}_i$, if $lift(\boldsymbol{z}, \boldsymbol{c}_i)_{[a,b]} > 0$. Here we regard $lift(\boldsymbol{z}, \boldsymbol{c}_i)$ as a matrix with the same shape of $\boldsymbol{z}$. This is achieved by averaging only over the timestep $t$ and not the feature dimension

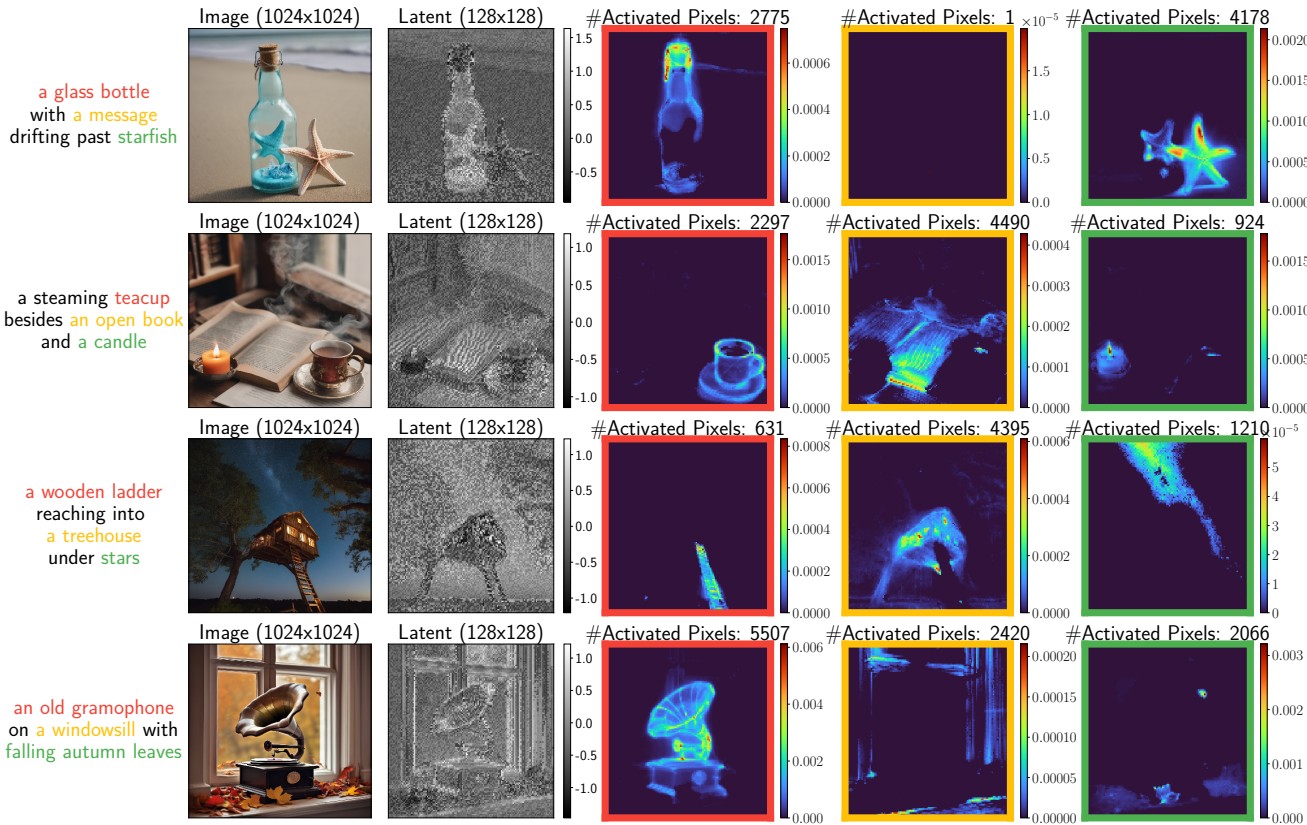

*Figure 7.* *CompLift* **for text-to-image compositional task.** Pixels with negative scores are masked out. From left to right: original image $x$, the latent $z$, the heatmaps of lift score $lift(z, c_i)$ for each object component $c_i$. The border color represents the corresponding object. We use random prompts and observe the clear pixel separation in general. Here the images with higher aesthetic qualities are presented.

when calculating the mean, i.e., Line 12 in Algorithm 1.

**Definition of *CompLift* in image space.** In particular, an object is considered to exist in the image $x$, if the number of activated pixels in its latent $z$ is greater than threshold $\tau$:

$$lift(x, c_i) := \sum_{a,b \in [1,m]} \mathbb{I}(lift(z, c_i)_{[a,b]} > 0) - \tau, \quad (4)$$

where $z$ is the latent in the shape of $(m, m)$. $\tau$ is a hyper-parameter to be tuned for our approach. We set $\tau = 250$ as the median activated pixel count among all images. Tests at 25th and 75th percentiles showed the median works best.

Equation (4) of *CompLift* for images makes the identification of objects more robust and interpretable. The rationale is that the activated pixels in the latent space are more likely to be the ones that correspond to the objects in the image. We present examples in Figure 7. As illustrated, we can see a clear consistency between the pixels activated by $lift(z, c_i)$ in the latent and the object $c_i$ in the image. Since the object often occupies only a fraction of the image, simply taking the mean along the feature dimension as Equation (3), may not be as effective as Equation (4), as those unrelated pixels with $lift(z, c_i) < 0$ will dominate the averaged score.

### 5.2. Reduce Variance of ELBO Estimation

Our initial attempt is to see whether the pixels in the latent $z$ align with the lift score. Intuitively, we expect for an arbitrary pixel with position $[a, b]$ from a target object $c_i$, its lift score $lift(z, c_i)_{[a,b]}$ should be greater than 0 in most cases. However, we found that the estimation variance is high when applying Equation (3) directly. As illustrated in the 3rd column in Figure 6, the lift score is noisy overall.

We hypothesize that the sampled noise $\epsilon$ in the L2 loss in Equation (3) is a major factor of the variance. In the ELBO estimation, we first add the noise to the image and make predictions, similar to adversarial attacks. The model may or may not have a good prediction on the new noisy image, therefore, the model's bias might lead to unstable estimation. To reduce the variance, we can find a candidate to replace $\epsilon$ when calculating the L2 loss in Equation (3). Ideally, the candidate is close enough to $\epsilon$, but inherently cancels out the bias from the model's prediction.

Our finding is that the composed score $\epsilon_\theta(z_t, c_{\text{compose}})$ is a good candidate to replace $\epsilon$, where $c_{\text{compose}}$ is the original composed prompt to generate latent $z_t$. For example, in Figure 6, $c_{\text{compose}}$ is "a black car and a white clock". Since

| Method | Product | | Mixture | | Negation | |
|---|---|---|---|---|---|---|
| | Acc ↑ | CD ↓ | Acc ↑ | CD ↓ | Acc ↑ | CD ↓ |
| Composable Diffusion | 56.5 | 0.061 | 70.7 | 0.042 | 7.9 | 0.207 |
| EBM (ULA) | 51.7 | 0.076 | 71.8 | 0.044 | 11.4 | 0.186 |
| EBM (U-HMC) | 56.3 | 0.036 | 73.1 | 0.063 | 9.4 | 0.269 |
| EBM (MALA) | 62.6 | 0.054 | 73.8 | 0.036 | 6.8 | 0.203 |
| EBM (HMC) | 64.8 | 0.021 | 73.7 | 0.078 | 4.2 | 0.340 |
| Composable Diffusion *+ Cached CompLift* | 99.5 | 0.009 | 86.5 | 0.023 | 62.0 | 0.137 |
| Composable Diffusion *+ CompLift* ($T = 50$) | 99.1 | 0.015 | 98.6 | 0.029 | 75.0 | 0.154 |
| Composable Diffusion *+ CompLift* ($T = 1000$) | **99.9** | **0.009** | **100.0** | **0.023** | **78.7** | **0.131** |

*Table 2.* **Quantitative results on 2D compositional generation.** Acc (%) means accuracy, and CD means Chamfer Distance.

$\epsilon_\theta(z_t, c_{\text{compose}})$ mainly guides the generation process of $z_t$, it is reasonable to assume that it is good at reconstructing the original latent, therefore is close to $\epsilon$. Furthermore, since $\epsilon_\theta(z_t, c_{\text{compose}})$ and $\epsilon_\theta(z_t, c_i)$ (or $\epsilon_\theta(z_t, \varnothing)$) come from the same model's prediction, it helps cancel the model's bias. The results are shown in the 2nd column of Figure 6. We observe that the variance is significantly reduced, and the activated pixels are more consistent with the objects.

The improved lift score for condition $c$ and latent $z$ is:

$$
\begin{aligned}
lift(z, c) := \mathbb{E}_{t,\epsilon} \{ &||\epsilon_\theta(z_t, c_{\text{compose}}) - \epsilon_\theta(z_t, \varnothing)||^2 \\
&- ||\epsilon_\theta(z_t, c_{\text{compose}}) - \epsilon_\theta(z_t, c)||^2 \}
\end{aligned}
\tag{5}
$$

Intuitively, if object $c$ exists in image $x$, then for most noisy latents $z_t$, $\epsilon_\theta(z_t, c)$ should be closer to $\epsilon_\theta(z_t, c_{\text{compose}})$ than the unconditional $\epsilon_\theta(z_t, \varnothing)$ in the corresponding pixels. The combination of Equations (4) and (5) represents our final criterion for text-to-image compositional task. We first estimate the lift score for each latent pixel via Equation (5), and then count the activated pixels to determine the existence of objects in the image via Equation (4).

## 6. Experiments

In the following sections, we evaluate the effectiveness of our *CompLift* criterion on 2D synthetic dataset, CLEVR Position dataset, and text-to-image compositional task.

### 6.1. 2D Synthetic Dataset

**Experiment setup.** We consider 2D synthetic dataset with 3 types of compositional algebra: product, mixture, and negation. We generate the distributions following the way in Du et. al (Du et al., 2023) - they are either Gaussian mixtures or uniform distribution. We use the same architecture of networks from (Du et al., 2023), which regards the diffusion model $\epsilon_\theta$ as $\nabla_x E_\theta$. $E_\theta$ is a 3-layer MLP and trained with the same loss function as the standard diffusion model by minimizing $\mathbb{E}_{t,\epsilon}||\nabla_x E_\theta(x_t, t) - \epsilon||_2^2$. We sample 8000 data

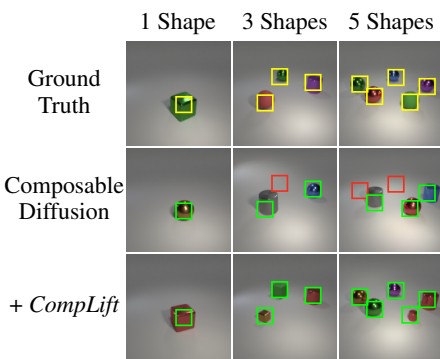

*Figure 8.* **Examples of samples on CLEVR Position dataset.**

points randomly for each component distribution, and train 1 individual diffusion model for each distribution. We use the diffusion process of 50 timesteps for both training and inference, which means the cached version of *CompLift* uses 50 trials. The vanilla version of *CompLift* uses 50 or 1000 trials. All the methods generate 8000 samples for inference. For *CompLift* algorithms, we use Composable Diffusion (Liu et al., 2022) to generate the initial 8000 samples, then apply the *CompLift* criterion to accept or reject samples. We test MCMC methods with Energy-Based Models (EBM) as additional baselines (Du et al., 2023), including Unadjusted Langevin Annealing (ULA), Unadjusted Hamiltonian Monte Carlo (U-HMC), Metropolis-Adjusted Langevin Algorithm (MALA) and Hamiltonian Monte Carlo (HMC). We provide more details in Appendix E.

**Metrics.** We define that a sample satisfies a uniform distribution if it falls into the nonzero-density regions. A sample satisfies Gaussian mixtures if it is within $3\sigma$ of any Gaussian component, where $\sigma$ is the standard deviation of the component. The negation condition is satisfied, if the above description is not satisfied. We calculate the accuracy of the algorithm on a composed distribution, as the percentage of generated samples that satisfy all the conditions. We use the Chamfer Distance to measure the similarity between the generated samples and the dataset, as we find it an efficient and general metric, applicable to all 3 kinds of algebra.

It is genuinely hard to define a universal $\epsilon_\theta(x_t, \varnothing)$ for the 2D synthetic task, since in this task, the 2D distribution can be of arbitrary shapes. Therefore, we find an effective strategy by defining $\epsilon_\theta(x_t, \varnothing) := \alpha \epsilon_\theta(x_t, c)$ with $\alpha = 0.9$ for each $c$. Consequently, the *lift* criterion is simplified as $\mathbb{E}_{t,\epsilon}\{||\epsilon - \alpha\epsilon_\theta(x_t, c)||^2 - ||\epsilon - \epsilon_\theta(x_t, c)||^2\}$. We find this criterion is surprisingly effective for this 2D task. One hypothesis is that if $x$ has some correlation with $c$, then a well-trained $\epsilon_\theta$ should be good at predicting the noise $\epsilon_\theta(x_t, c)$ that is closer to $\epsilon$ and further from $\mathbf{0}$ in most cases.

The overall result is shown in Table 2. We observe that the *CompLift* criterion is effective in all 3 types of compositional algebra. Compared to MCMC methods, rejection

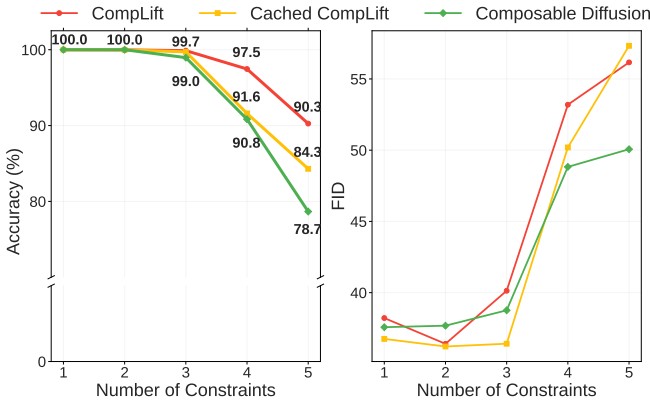

*Figure 9.* **Quantitative results on CLEVR Position dataset.**

| Method | Animals | | Object&Animal | | Objects | |
|---|---|---|---|---|---|---|
| | CLIP ↑ | IR ↑ | CLIP ↑ | IR ↑ | CLIP ↑ | IR ↑ |
| Stable Diffusion 1.4 | 0.310 | -0.191 | 0.343 | 0.432 | 0.333 | -0.684 |
| SD 1.4 + EBM (ULA) | 0.311 | 0.026 | 0.342 | 0.387 | 0.344 | -0.380 |
| SD 1.4 + *Cached CompLift* | 0.319 | 0.128 | 0.356 | 0.990 | 0.344 | -0.131 |
| SD 1.4 + *CompLift (T=50)* | 0.320 | 0.241 | 0.355 | 0.987 | 0.344 | -0.154 |
| SD 1.4 + *CompLift (T=200)* | **0.322** | **0.293** | **0.358** | **1.093** | **0.347** | **-0.050** |
| Stable Diffusion 2.1 | 0.330 | 0.532 | 0.354 | 0.924 | 0.342 | -0.112 |
| SD 2.1 + EBM (ULA) | 0.330 | 0.829 | 0.357 | 0.981 | 0.348 | 0.218 |
| SD 2.1 + *Cached CompLift* | 0.339 | 0.880 | 0.361 | 1.252 | 0.354 | 0.353 |
| SD 2.1 + *CompLift (T=50)* | 0.340 | **0.992** | 0.361 | 1.263 | 0.354 | 0.454 |
| SD 2.1 + *CompLift (T=200)* | **0.340** | 0.975 | **0.362** | **1.283** | **0.355** | **0.489** |
| Stable Diffusion XL | 0.338 | 1.025 | 0.363 | 1.621 | 0.359 | 0.662 |
| SD XL + EBM (ULA) | 0.335 | 0.913 | 0.362 | 1.676 | 0.361 | 0.872 |
| SD XL + *Cached CompLift* | 0.341 | **1.244** | **0.364** | 1.687 | 0.365 | **0.896** |
| SD XL + *CompLift (T=50)* | 0.342 | 1.222 | 0.364 | 1.700 | 0.365 | 0.842 |
| SD XL + *CompLift (T=200)* | **0.342** | 1.216 | **0.364** | **1.706** | **0.367** | 0.890 |

*Table 3.* **Quantitative results on Attend-and-Excite Benchmarks (Chefer et al., 2023).** IR is ImageReward (Xu et al., 2024).

with *CompLift* criterion is able to generate empty sets by construction. We show some examples in Figure 1, and more examples are illustrated in Appendix E.

### 6.2. CLEVR Position Dataset

**Experiment setup.** The CLEVR Position dataset (Johnson et al., 2017) is a dataset of rendered images with a variety of objects placed in different positions. Given a set of 1-5 specified positions, the model needs to generate an image that contains all objects in the specified positions (i.e., product algebra). We use the pretrained diffusion model from (Liu et al., 2022). The model is trained with importance sampling (Nichol & Dhariwal, 2021). Thus, we use importance sampling for ELBO estimation. In particular, we find that sampling $t$ solely as $t = 928$ is sufficient to give us a decent performance by observing Figure 3. This leads to a computational efficiency via only 1 trial at $t = 928$ for both vanilla and cached versions of *CompLift*. For each composition of conditions, our algorithm uses Composable Diffusion (Liu et al., 2022) to generate the initial 10 samples, then apply the *Lift* criterion to accept or reject samples. We test all methods with 5000 combinations of positions with various numbers of constraints. All methods use classifier-free guidance with a guidance scale of 7.5 (Ho & Salimans, 2022).

**Metrics.** We use Segment Anything Model 2 (SAM2) (Ravi et al., 2024) as a generalizable verifier, to check whether an object is in a specified position. We first extract the background mask, which is the mask with the largest area, using the automatic mask generator of SAM2, then label a condition as satisfied if the targeted position is not in the background mask. We find this new verifier is more robust compared to the pretrained classifier in Liu et al. (2022). We provide more details in Appendix F. We calculate the accuracy as the ratio of samples that satisfy all given conditions. To calculate FID, we compare the generated samples to the training set provided by Liu et al. (2022).

We show the quantitative results in Figure 9. We observe

that the *CompLift* criterion is able to generate samples that satisfy all conditions more effectively than the baseline. As the number of constraints increases, the performance of the baseline drops significantly, while the performance of the *CompLift* criterion remains relatively stable. We also show some examples in Figure 8. Meanwhile, we observe some regressions in the FID scores. We hypothesize that the rejection reduces the diversity of samples, which results in the higher FID scores, since the image quality has no significant difference from Composable Diffusion.

### 6.3. Text-to-Image Compositional Task

**Experiment setup.** We use the pretrained Stable Diffusion 1.4, 2.1, and SDXL (Rombach et al., 2022; Podell et al., 2023). The models are trained with large-scale text-to-image dataset LAION (Schuhmann et al., 2022). We use the same composed prompts as Attend-and-Excite Benchmark (Chefer et al., 2023), which considers 3 classes of compositions - 2 animal categories, 1 animal and 1 object category, and 2 object categories (i.e., product algebra). There are 66 unique animal combinations, 66 unique object combinations, and 144 unique animal-object combinations. We use the diffusion model to generate 5 initial samples using each combined prompt, then apply the *CompLift* criterion to accept or reject samples. For inference, we use 50 diffusion timesteps, which means the cached version of *CompLift* uses 50 trials. The vanilla version of *CompLift* uses 200 trials. We use CFG with a guidance scale of 7.5 for all methods. We fix $\tau = 250$ for all experiments.

**Metrics.** We use CLIP (Radford et al., 2021) and ImageReward (Xu et al., 2024) to assess the alignment between the generated images and the prompts. For combination where *CompLift* is invalid for all generated samples, we uniformly sample one of the images as the final output. The metrics are averaged over the samples within each combination class.

The quantitative results are shown in Table 3. We observe that with *CompLift*, the performance for each version of SD

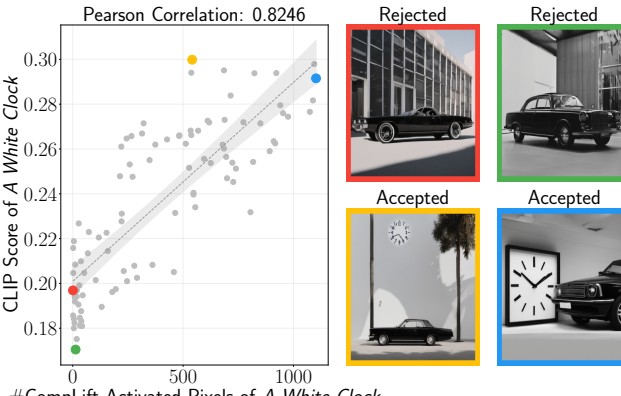

*Figure 10.* **The correlation between the number of activated pixels and the CLIP score.** The correlation is calculated using the Pearson correlation coefficient. Each point represents a generated image. Prompt: *a black car and a white clock.*

can be boosted to be comparable to their next generation for some combination classes. For example, SD 1.4 with *CompLift* is comparable to the vanilla SD 2.1 on object-animal combinations and object combinations. SD 2.1 with *CompLift* is comparable to the vanilla SDXL on animal combinations. Since the *CompLift* criterion is a training-free criterion and does not depend on additional verifier, it also shows a novel way of how diffusion model could self-improve during test time without extra training cost (Ma et al., 2025). Additionally, we show its correlation with CLIP in Figure 10, and leave the discussion in Appendix G.

## 7. Conclusion

We propose a novel rejection criterion *CompLift* for compositional tasks, which is based on the difference of the ELBO between the generated samples with and without the given conditions. We show that the *CompLift* criterion is effective on a 2D synthetic dataset, CLEVR Position dataset, and text-to-image compositional task. We also demonstrate that the *CompLift* criterion can be further optimized with caching strategies.

Some limitations include that the performance of *CompLift* is dependent on the quality of the underlying generative model. For example, if the model is not well-trained, it may not be able to generate any acceptable samples. Another limitation is that while we tested all algebras in the 2D dataset, we did not have a sysmatic test on OR/NOT algebras for text-to-image generation due to a lack of existing mature benchmarks for OR/NOT algebras. It would also be interesting to use segmentation models like Grounded Segment Anything (Ren et al., 2024) to provide a grounded evaluation of the accuracy of the *CompLift* criterion.

In the future, we plan to explore the potential of the *CompLift* criterion on more complex compositional tasks, such as video generation and music generation.

## Acknowledgements

We thank the anonymous reviewers for their helpful comments in revising the paper. This material is based on work supported by NSF Career CCF 2047034, NSF CCF DASS 2217723, and NSF AI Institute CCF 2112665.

## Impact Statement

We introduce *CompLift*, a novel resampling criterion leveraging lift scores to advance the compositional capabilities of diffusion models. While we mainly focus on efficient approximation of lift scores without additional training and improving compositional generation, we acknowledge several broader implications, similar to other works on diffusion models and image generation.

As we demonstrate in this work, *CompLift* could benefit image generation applications through more reliable generation of complex, multi-attribute outputs. On the positive side, *CompLift* could enhance creative tools through more precise and reliable image generation, potentially enabling more sophisticated applications in art, design, and entertainment. However, like other works on diffusion models, *CompLift* could potentially be misused to create misleading or deepfake contents, if not properly managed (Mirsky & Lee, 2021). These contents could be more consistent with the user intention, leading to potential societal harms such as scams and misinformation. Additionally, biases in training data could be amplified through the resampling process, potentially perpetuating societal biases and underrepresentation in the generated images. We encourage work building on *CompLift* to evaluate not only technical performance but also the risks and ethical implications of the approach.

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

## A. Connection with Classifier-Free Guidance (CFG) (Ho & Salimans, 2022; Liu et al., 2022)

A brief summary is that our method tries to solve the primal form as Equation (6), while CFG tries to solve the dual form using soft Lagrangian regularization to satisfy the constraint as Equation (9). The two formulations are equivalent in the Lagrangian sense, but CFG is not guaranteed to strictly satisfy the constraints in practice. Though in practice, we will lose the theoretical guarantee of *CompLift* using ELBO estimation, but we hope this theoretical perspective can help the readers understand how *CompLift* aims to directly solve the primal form.

We provide a rough proof as follows. The formulation of the final probability can be regarded as a constrained distribution:

$$\boldsymbol{x}_0 \sim p_{\text{generator}}(\boldsymbol{x}_0), \quad \text{s.t. } p(\boldsymbol{x}_0 \mid \boldsymbol{c}_i) > p(\boldsymbol{x}_0), \quad \forall \boldsymbol{c}_i. \tag{6}$$

Now we show that CFG (Ho & Salimans, 2022), or Composable Diffusion (which uses CFG in the compositional generation context) (Liu et al., 2022), try to satisfy the constraint using soft regularization. For convenience, we take the logarithm (assuming all probabilities are strictly positive). Ignoring the boundary condition, this transforms the constraint $p(\boldsymbol{x}_0 \mid \boldsymbol{c}_i) > p(\boldsymbol{x}_0)$ into:

$$\log p(\boldsymbol{x}_0 \mid \boldsymbol{c}_i) - \log p(\boldsymbol{x}_0) \geq 0, \quad \forall \boldsymbol{c}_i. \tag{7}$$

Suppose our goal is to maximize the log-likelihood of the generator while ensuring the above constraints are met. The optimization problem can be written as:

$$\max_{\boldsymbol{x}_0} \quad \log p_{\text{generator}}(\boldsymbol{x}_0)$$
$$\text{subject to} \quad \log p(\boldsymbol{x}_0 \mid \boldsymbol{c}_i) - \log p(\boldsymbol{x}_0) \geq 0, \quad \forall \boldsymbol{c}_i. \tag{8}$$

To handle the constraints, we introduce non-negative Lagrange multipliers $\lambda_i \geq 0$ for each constraint. The Lagrangian function is then defined as:

$$\boxed{\mathcal{L}(\boldsymbol{x}_0, \lambda) = \log p_{\text{generator}}(\boldsymbol{x}_0) + \sum_{\boldsymbol{c}_i} \lambda_i \Big( \log p(\boldsymbol{x}_0 \mid \boldsymbol{c}_i) - \log p(\boldsymbol{x}_0) \Big), \quad \lambda_i \geq 0.} \tag{9}$$

In this formulation:

- The first term, $\log p_{\text{generator}}(\boldsymbol{x}_0)$, is our original objective.

- The second term penalizes any violation of the constraint $\log p(\boldsymbol{x}_0 \mid \boldsymbol{c}_i) - \log p(\boldsymbol{x}_0) \geq 0$. If the constraint is violated, the penalty term will lower the overall value of the Lagrangian.

We assume the derivative of $\log p_\theta(\boldsymbol{x}_0)$ at $\boldsymbol{x}_t$ is proportional to $\epsilon_\theta(\boldsymbol{x}_t, \varnothing)$:

$$\nabla_{\boldsymbol{x}_t} \log p_\theta(\boldsymbol{x}_0) \approx \nabla_{\boldsymbol{x}_t} \mathbb{E} \epsilon, t(\epsilon\theta(\boldsymbol{x}_t, \varnothing) - \epsilon)_2^2 \propto \epsilon_\theta(\boldsymbol{x}_t, \varnothing). \tag{10}$$

Consequently, assuming that $p_{\text{generator}}(\boldsymbol{x}_0) \approx p_\theta(\boldsymbol{x}_0, \varnothing)$, then the derivative of the Lagrangian at $\boldsymbol{x}_t$:

$$\boxed{\nabla_{\boldsymbol{x}_t} \mathcal{L}(\boldsymbol{x}_0, \lambda) \propto \epsilon_\theta(\boldsymbol{x}_t, \varnothing) + \sum_{\boldsymbol{c}_i} \lambda_i \Big( \epsilon_\theta(\boldsymbol{x}_t, \boldsymbol{c}_i) - \epsilon_\theta(\boldsymbol{x}_t, \varnothing) \Big), \quad \lambda_i \geq 0,} \tag{11}$$

which derives the exact CFG-like formulation as Equation 11 in Composable Diffusion (Liu et al., 2022).

Some notes:

- CFG / Composable Diffusion fix the Lagrangian coefficient $\lambda_i$ as a fixed weight value $w$. Yet, it is unknown whether the constraints are satisfied indeed.

- In text-to-image tasks, *CompLift* uses $p_{\text{generator}}(\boldsymbol{x}_0)$ as $p_\theta(\boldsymbol{x}_0, \boldsymbol{c}_{\text{compose}})$, while CFG / Composable Diffusion use $p_{\text{generator}}(\boldsymbol{x}_0)$ as $p_\theta(\boldsymbol{x}_0, \varnothing)$. Using $p_\theta(\boldsymbol{x}_0, \boldsymbol{c}_{\text{compose}})$ in *CompLift* increases the probability of the samples to be accepted in practice. It would be interesting to test whether changing the $p_{\text{generator}}(\boldsymbol{x}_0)$ to $p_\theta(\boldsymbol{x}_0, \boldsymbol{c}_{\text{compose}})$ in CFG / Composable Diffusion can also improve the performance.

## B. `Compose` Algorithm

We describe the `Compose` algorithm used in our experiments in Algorithm 3. The algorithm takes as input a set of lift scores $\{lift(\boldsymbol{x}_0|\boldsymbol{c}_i)\}_{i=1}^n$ and an algebra $\mathcal{A}$, and returns a boolean value indicating whether the composed prompt is satisfied. The algorithm first initializes a dictionary $s$ to store partial results. It then converts the algebra $\mathcal{A}$ to conjunctive normal form (CNF) and iterates over each disjunction $\mathcal{A}_k$ in $\mathcal{A}$. For each literal $\mathcal{L}_m$ in $\mathcal{A}_k$, the algorithm calculates the lift score based on the input lift scores and updates $s[\mathcal{A}_k]$. Finally, it returns whether the minimum of the results is greater than zero.

---

**Algorithm 3** `Compose` for multiple algebraic operations

---

1: **Input:** lift scores $\{lift(\boldsymbol{x}_0|\boldsymbol{c}_i)\}_{i=1}^n$, algebra $\mathcal{A}$
2: Initialize $s = \{\}$                                                   ▷ Dictionary to store partial results
3: Convert $\mathcal{A}$ to conjunctive normal form (CNF)
4: **for** disjunction $\mathcal{A}_k \in \mathcal{A}$ **do**
5:     Initialize $s[\mathcal{A}_k] = 0$
6:     **for** literal $\mathcal{L}_m \in \mathcal{A}_k$ **do**
7:         **if** $\mathcal{L}_m$ is of form $\boldsymbol{c}_i$ **then**
8:             $s[\mathcal{A}_k] = \max(lift(\boldsymbol{x}_0|\boldsymbol{c}_i), s[\mathcal{A}_k])$
9:         **else if** $\mathcal{L}_m$ is of form $\neg\boldsymbol{c}_i$ **then**
10:            $s[\mathcal{A}_k] = \max(-lift(\boldsymbol{x}_0|\boldsymbol{c}_i), s[\mathcal{A}_k])$
11:         **end if**
12:     **end for**
13: **end for**
14: **return** $\min_{\mathcal{A}_k \in \mathcal{A}}(s[\mathcal{A}_k]) > 0$

---

## C. *CompLift* for Text-to-Image Generation (Vanilla Version)

We combine Algorithm 1, Equation (4), and Equation (5), and provide the full algorithm of *CompLift* for t2i tasks.

---

**Algorithm 4** Text-to-Image Generation with CompLift

---

1: **Input:** image $\boldsymbol{x}_0$, conditions $\{\boldsymbol{c}_i\}_{i=1}^n$, composed prompt $\boldsymbol{c}_{\text{compose}}$, # of trials $T$, threshold $\tau$
2: Initialize `PixelLift`$[\boldsymbol{c}_i]$ = list() for each $\boldsymbol{c}_i$
3: Encode $\boldsymbol{x}_0$ to latent $\boldsymbol{z}_0$ using VAE encoder
4: **for** trial $j = 1, \ldots, T$ **do**
5:     Sample $t \sim [1, 1000]$; $\epsilon \sim \mathcal{N}(0, I)$
6:     $\boldsymbol{z}_t = \sqrt{\bar{\alpha}_t}\boldsymbol{z}_0 + \sqrt{1 - \bar{\alpha}_t}\epsilon$
7:     $\epsilon_{\text{compose}} = \epsilon_\theta(\boldsymbol{z}_t, \boldsymbol{c}_{\text{compose}})$                                  ▷ Get composed prediction
8:     **for** condition $\boldsymbol{c}_k \in \{\boldsymbol{c}_i\}_{i=1}^n$ **do**
9:         $lift_j(\boldsymbol{z}_0|\boldsymbol{c}_k) = \|\epsilon_{\text{compose}} - \epsilon_\theta(\boldsymbol{z}_t, \varnothing)\|^2 - \|\epsilon_{\text{compose}} - \epsilon_\theta(\boldsymbol{z}_t, \boldsymbol{c}_k)\|^2$
10:         `PixelLift`$[\boldsymbol{c}_k]$.append$(lift_j(\boldsymbol{z}_0|\boldsymbol{c}_k))$
11:     **end for**
12: **end for**
13: **for** condition $\boldsymbol{c}_k \in \{\boldsymbol{c}_i\}_{i=1}^n$ **do**
14:     $lift(\boldsymbol{z}_0, \boldsymbol{c}_k) = \text{mean}(\texttt{PixelLift}[\boldsymbol{c}_k])$                         ▷ Average over trials
15:     $lift(\boldsymbol{x}_0, \boldsymbol{c}_k) = \sum\limits_{a,b \in [1,m]} \mathbb{I}(lift(\boldsymbol{z}_0, \boldsymbol{c}_k)_{[a,b]} > 0) - \tau$
16: **end for**
17: **return** `Compose`$(\{lift(\boldsymbol{x}_0|\boldsymbol{c}_i)\}_{i=1}^n, \mathcal{A})$                             ▷ Run Algorithm 3

---

## D. *Cached CompLift* for Text-to-Image Generation (Cached Version)

Similar to the vanilla version, we combine Algorithm 2, Equation (4), and Equation (5), and provide the full algorithm of *Cached CompLift* for t2i tasks. The algorithm is divided into two parts: Algorithm 5 and Algorithm 6. The first part caches the latent vectors and predictions at each timestep, while the second part calculates the lift scores using the cached values.

---

**Algorithm 5** Cache Values During Text-to-Image Generation

---

1: **Input:** conditions $\{c_i\}_{i=1}^n$, composed prompt $c_{\text{compose}}$, # of timesteps $N$
2: Initialize $z_T \sim \mathcal{N}(0, I)$
3: Initialize Cache $= \{\}$          ▷ Dictionary to store intermediate values
4: **for** timestep $t_j \in [t_N, t_{N-1}, \dots, t_1]$ **do**
5:     ▷ Standard generation process
6:     $z_{t_{j-1}} = \text{step}(z_{t_j}, \epsilon_\theta, c_{\text{compose}})$
7:     ▷ Add predictions to cache
8:     Cache$[t_j]$.latent $= z_{t_j}$
9:     Cache$[t_j]$.null_pred $= \epsilon_\theta(z_{t_j}, \varnothing)$
10:     **for** condition $c_k \in \{c_i\}_{i=1}^n$ **do**
11:        Cache$[t_j]$.cond_pred$[c_k] = \epsilon_\theta(z_{t_j}, c_k)$
12:     **end for**
13:     Cache$[t_j]$.compose_pred $= \epsilon_\theta(z_{t_j}, c_{\text{compose}})$
14: **end for**
15: Decode $z_0$ to image $x_0$ using VAE decoder
16: **return** $x_0$, Cache

---

**Algorithm 6** CompLift Using Cached Values

---

1: **Input:** image $x_0$, conditions $\{c_i\}_{i=1}^n$, Cache, threshold $\tau$
2: Initialize PixelLift$[c_i] = \text{list}()$ for each $c_i$
3: Encode $x_0$ to latent $z_0$ using VAE encoder
4: **for** timestep $t_j$ in Cache **do**
5:     $z_{t_j} = \text{Cache}[t_j]$.latent
6:     $\epsilon_{\text{compose}} = \text{Cache}[t_j]$.compose_pred
7:     **for** condition $c_k \in \{c_i\}_{i=1}^n$ **do**
8:        $lift_j(z_0|c_k) = \|\epsilon_{\text{compose}} - \text{Cache}[t_j].\text{null\_pred}\|^2 - \|\epsilon_{\text{compose}} - \text{Cache}[t_j].\text{cond\_pred}[c_k]\|^2$
9:        PixelLift$[c_k]$.append($lift_j(z_0|c_k)$)
10:     **end for**
11: **end for**
12: **for** condition $c_k \in \{c_i\}_{i=1}^n$ **do**
13:     $lift(z_0, c_k) = \text{mean}(\text{PixelLift}[c_k])$          ▷ Average over timesteps
14:     $lift(x_0, c_k) = \sum\limits_{a,b \in [1,m]} \mathbb{I}(lift(z_0, c_k)_{[a,b]} > 0) - \tau$
15: **end for**
16: **return** Compose($\{lift(x_0|c_i)\}_{i=1}^n, \mathcal{A}$)          ▷ Run Algorithm 3

---

# E. 2D Synthetic Distribution Compositions

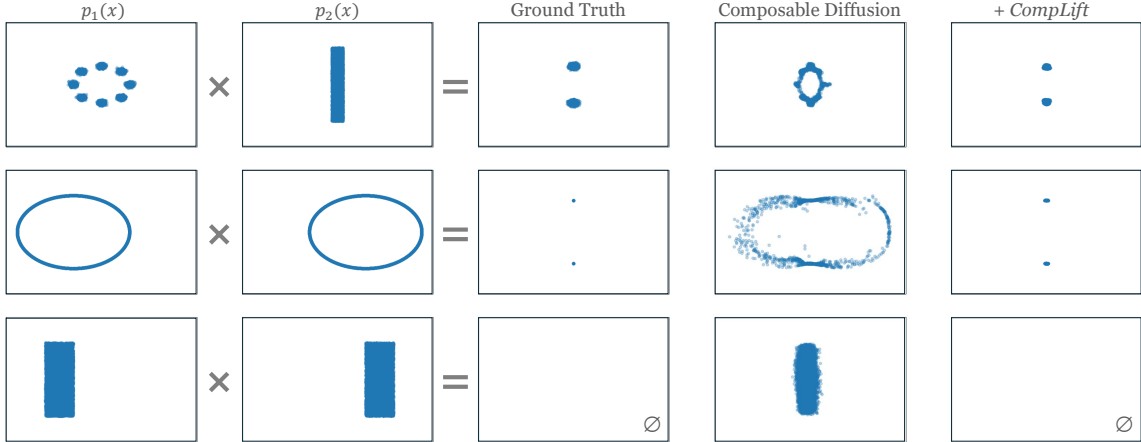

*Figure 11.* 2D synthetic result for product composition with Composable Diffusion and *CompLift*.

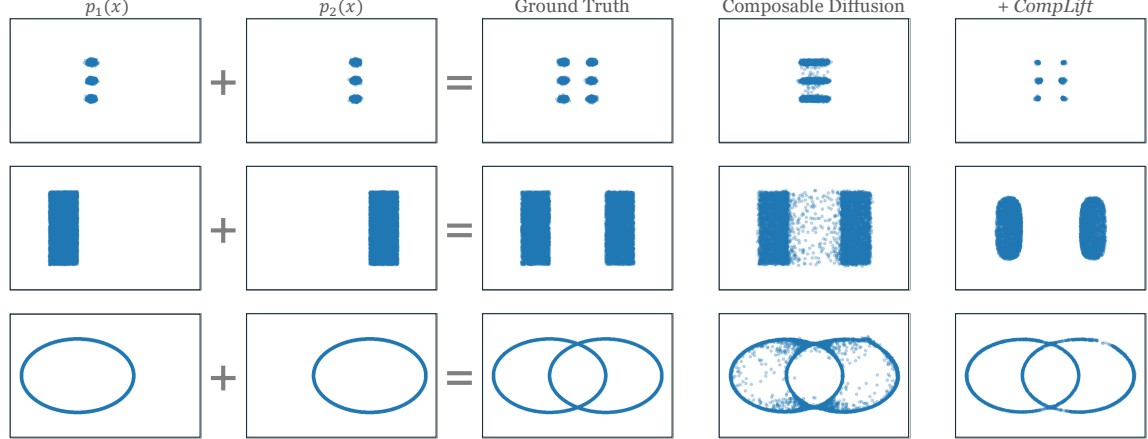

*Figure 12.* 2D synthetic result for mixture composition with Composable Diffusion and *CompLift*.

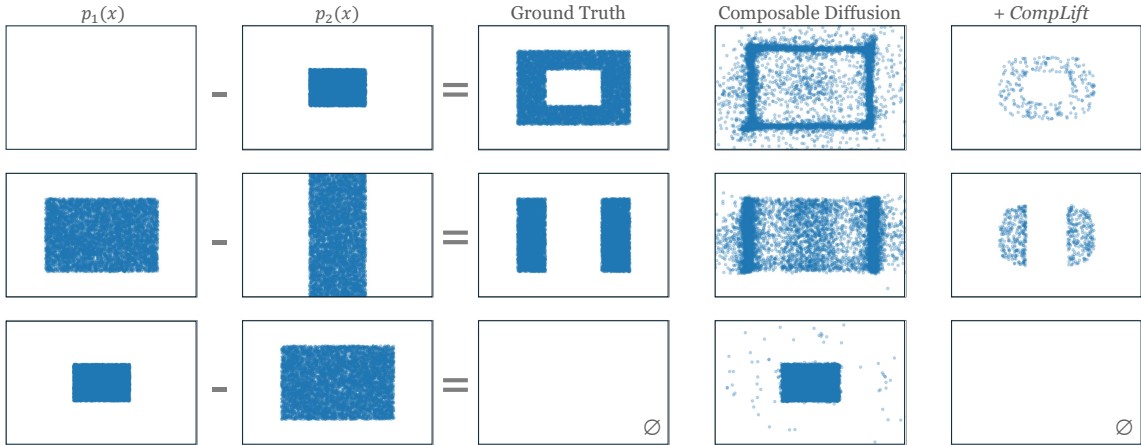

*Figure 13.* 2D synthetic result for negation composition with Composable Diffusion and *CompLift*.

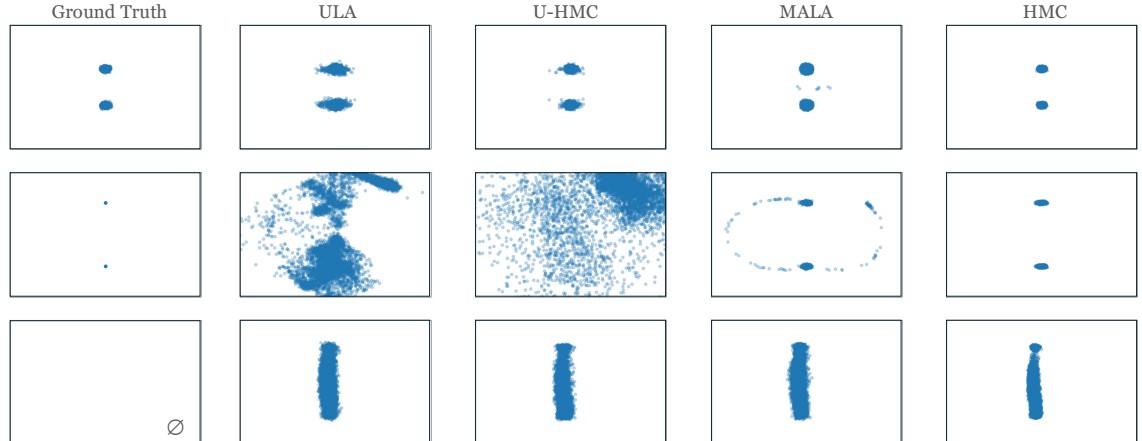

*Figure 14.* 2D synthetic result for product composition with Energy-Based Models and MCMC. The component distributions are the same ones in Figure 11.

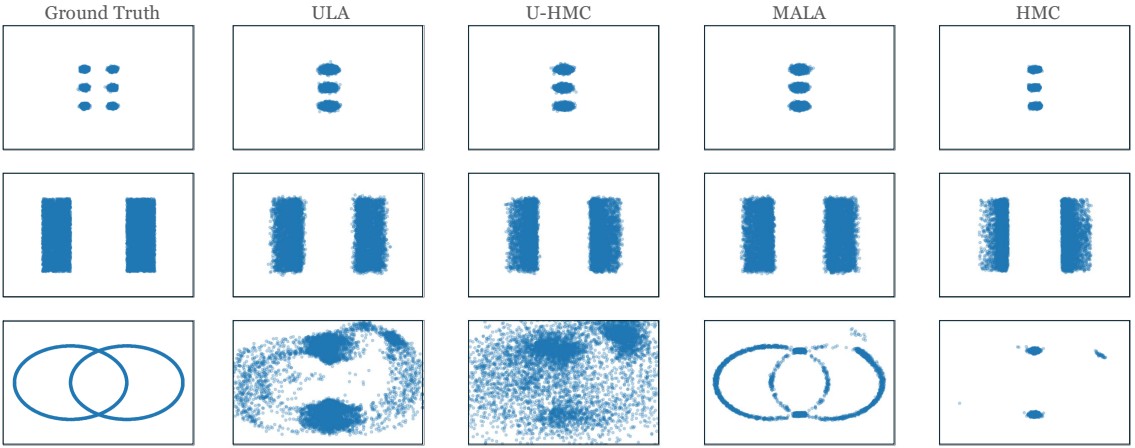

*Figure 15.* 2D synthetic result for mixture composition with Energy-Based Models and MCMC. The component distributions are the same ones in Figure 12.

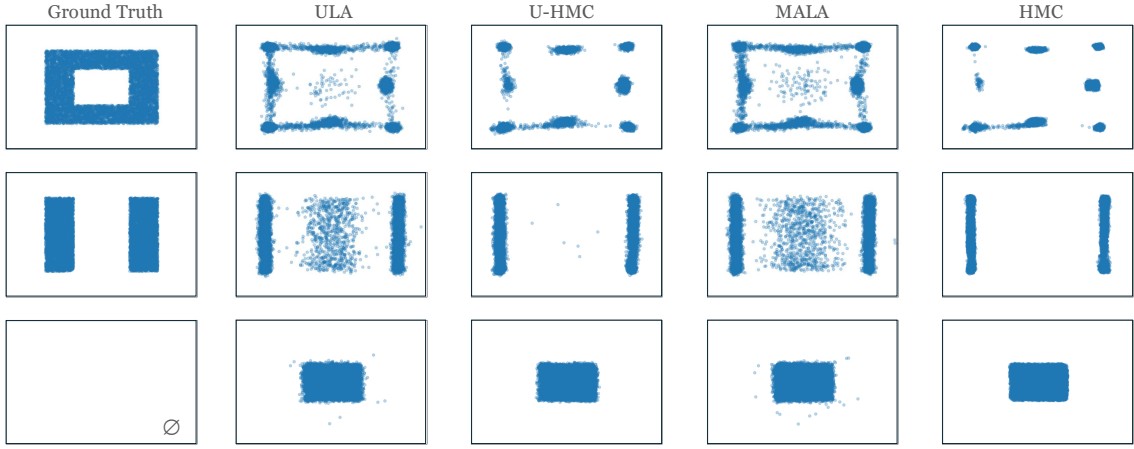

*Figure 16.* 2D synthetic result for negation composition with Energy-Based Models and MCMC. The component distributions are the same ones in Figure 13.

In Figures 11 to 16, we provide additional visualizations of the 2D synthetic distribution compositions. The figures show the generated samples for product, mixture, and negation compositions, respectively. We find that the summation result is slightly different from the visualization result in Du et. al (Du et al., 2023), which is likely due to the difference of the temperature used for the composed distribution. We simply set temperature as the default 1 for the composed distribution of the summation algebra, while Du et. al (Du et al., 2023) sets a temperature as 3.5 (see the implementation of MixtureEBMDiffusionModel in `https://github.com/yilundu/reduce_reuse_recycle/blob/main/notebooks/simple_distributions.ipynb`). We think our choice may better represent what the original composed distribution can achieve in practice.

Table 4 shows the detailed results for the 2D synthetic distribution compositions. We provide the accuracy (Acc), Chamfer distance (CD), and acceptance ratio (Ratio) of samples using the *CompLift* algorithm for each composition. We find that Composable Diffusion enhanced with *CompLift* ($T = 1000$) consistently outperforms baseline methods across all scenarios, achieving perfect or near-perfect accuracy in many cases. While all methods perform reasonably well with Product algebra, performance generally degrades with Mixture and especially with Negation algebra, where *CompLift* still maintains a significant edge over other approaches. The results demonstrate that *CompLift* effectively improves both accuracy and sample quality (measured by Chamfer Distance), though this comes with varying sampling efficiency as shown by the acceptance ratios.

Figure 17 shows the histograms of the *CompLift* scores for each *composed* 2D synthetic distributions following the order in the Figure 11, Figure 12, and Figure 13, where the groud-truth positive data points are in color blue, and the negative data points are in color orange. We observe that ❶ positive data samples have a higher *CompLift* score and negative data samples (i.e., the data sample out of the desired distribution) have a lower *CompLift* score, and ❷ the boundary condition of 0 is a good threshold to separate the positive and negative data samples for product. For mixture and negation, the 0 boundary condition is less effective and slightly conservative, but the positive data samples still have a higher *CompLift* score than the negative data samples, and the performance still achieves better when applying *CompLift* to the baseline methods.

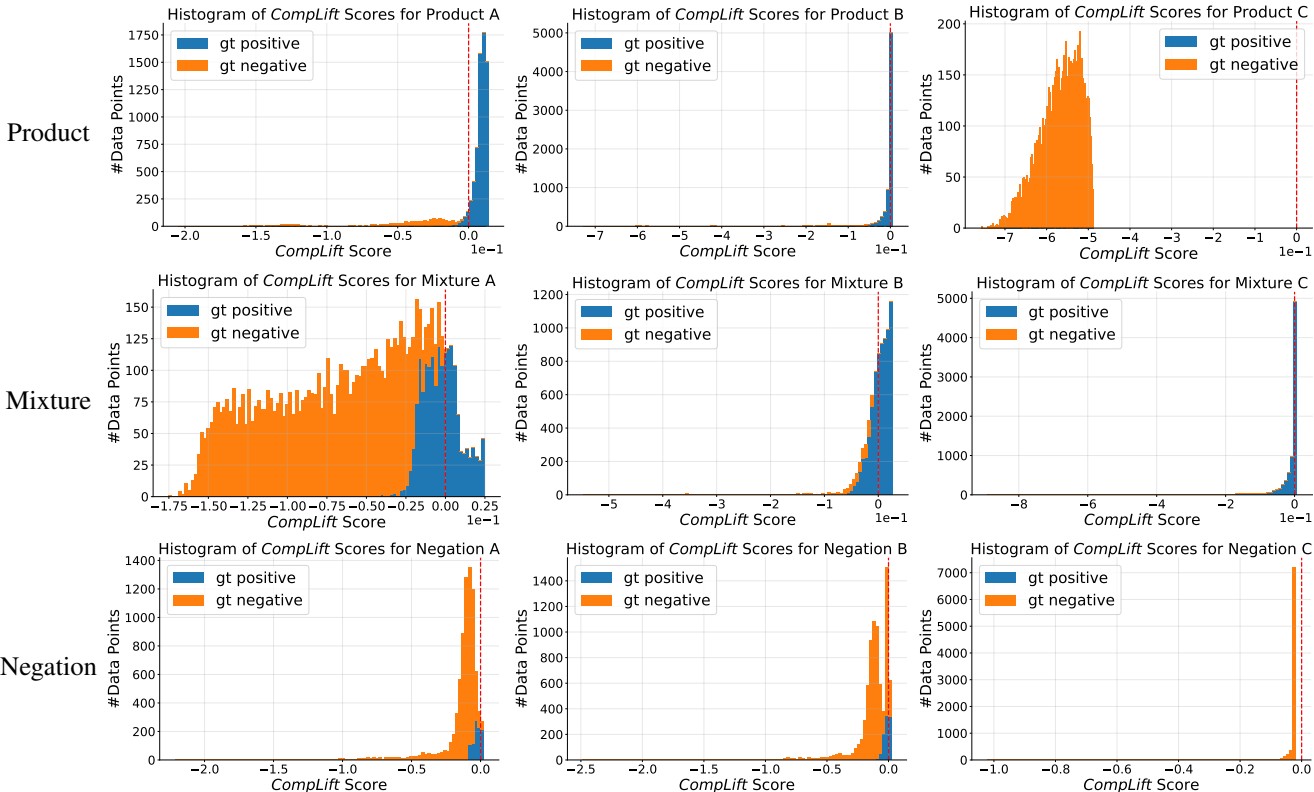

*Figure 17.* **Histograms of the *CompLift* scores for 2D synthetic distribution compositions.** The histograms show the distribution of the lift scores for the three compositions. The samples are 8000 samples generated from Composable Diffusion. The x-axis represents the composed lift scores using Table 1, and the y-axis represents the number of samples.

| Algebra | Method | Scenario 1 | | | Scenario 2 | | | Scenario 3 | | |
|---|---|---|---|---|---|---|---|---|---|---|
| | | Acc ↑ | CD ↓ | Ratio | Acc ↑ | CD ↓ | Ratio | Acc ↑ | CD ↓ | Ratio |
| Product | Composable Diffusion | 81.9 | 0.030 | - | 87.6 | 0.093 | - | 0.0 | - | - |
| | EBM (ULA) | 80.3 | 0.010 | - | 74.7 | 0.141 | - | 0.0 | - | - |
| | EBM (U-HMC) | 84.9 | 0.007 | - | 83.9 | 0.065 | - | 0.0 | - | - |
| | EBM (MALA) | 95.8 | 0.005 | - | 92.0 | 0.103 | - | 0.0 | - | - |
| | EBM (HMC) | 95.7 | **0.003** | - | 98.8 | 0.039 | - | 0.0 | - | - |
| | Composable Diffusion + *Cached CompLift* | 98.5 | 0.005 | - | **100.0** | 0.013 | - | **100.0** | - | - |
| | Composable Diffusion + *CompLift* ($T = 50$) | 97.4 | 0.011 | - | **100.0** | 0.019 | - | **100.0** | - | - |
| | Composable Diffusion + *CompLift* ($T = 1000$) | **99.8** | 0.005 | 79.3 | **100.0** | **0.013** | 55.1 | **100.0** | - | 0.0 |
| Mixture | Composable Diffusion | 22.1 | 0.067 | - | 90.0 | 0.025 | - | 99.9 | 0.033 | - |
| | EBM (ULA) | 28.2 | 0.062 | - | 87.9 | 0.021 | - | 99.2 | 0.049 | - |
| | EBM (U-HMC) | 30.5 | 0.060 | - | 89.5 | 0.020 | - | 99.5 | 0.108 | - |
| | EBM (MALA) | 28.4 | 0.059 | - | 92.9 | **0.018** | - | 100.0 | 0.030 | - |
| | EBM (HMC) | 28.7 | 0.059 | - | 92.3 | 0.019 | - | 100.0 | 0.157 | - |
| | Composable Diffusion + *Cached CompLift* | 61.0 | 0.030 | - | 98.5 | 0.022 | - | **100.0** | 0.018 | - |
| | Composable Diffusion + *CompLift* ($T = 50$) | 95.7 | 0.020 | - | **100.0** | 0.041 | - | **100.0** | 0.026 | - |
| | Composable Diffusion + *CompLift* ($T = 1000$) | **100.0** | **0.017** | 9.2 | **100.0** | 0.038 | 56.8 | **100.0** | **0.014** | 57.1 |
| Negation | Composable Diffusion | 11.8 | 0.191 | - | 11.7 | 0.222 | - | 0.0 | - | - |
| | EBM (ULA) | 28.7 | 0.129 | - | 5.6 | 0.244 | - | 0.0 | - | - |
| | EBM (U-HMC) | 25.9 | 0.206 | - | 2.3 | 0.333 | - | 0.0 | - | - |
| | EBM (MALA) | 17.8 | 0.150 | - | 2.7 | 0.257 | - | 0.0 | - | - |
| | EBM (HMC) | 12.4 | 0.258 | - | 0.2 | 0.421 | - | 0.0 | - | - |
| | Composable Diffusion + *Cached CompLift* | 40.4 | 0.152 | - | 45.6 | **0.123** | - | **100.0** | - | - |
| | Composable Diffusion + *CompLift* ($T = 50$) | 68.6 | 0.150 | - | 56.4 | 0.158 | - | **100.0** | - | - |
| | Composable Diffusion + *CompLift* ($T = 1000$) | **78.7** | **0.127** | 3.2 | **57.2** | 0.135 | 6.3 | **100.0** | - | 0.0 |

*Table 4.* Quantitative results on 2D composition. Acc means accuracy, CD means Chamfer Distance, and Ratio means the acceptance ratio of samples using *CompLift*.

### E.1. Parameters of Distributions

The distributions follow the generation way in Du et. al (Du et al., 2023) - they are either Gaussian mixtures or uniform distribution. We sample 8000 data points randomly for each component distribution, and train 1 diffusion model for each distribution. We define that a sample satisfies a uniform distribution if it falls into the nonzero-density regions. A sample satisfies Gaussian mixtures if it is within $3\sigma$ of any Gaussian component, where $\sigma$ is the standard deviation of the component. The negation condition is satisfied, if the above description is not satisfied.

- **Product, Component A1:** A mixture of 8 Gaussians evenly placed on a circle of radius $0.5$ centered at the origin. Each Gaussian has a standard deviation of $0.3$.

- **Product, Component A2:** A uniform distribution in a vertical strip: $x \in [-0.1, 0.1]$, $y \in [-1, 1]$.

- **Product, A1 $\wedge$ A2:** Two Gaussian blobs centered at $(0, 0.5)$ and $(0, -0.5)$, each with standard deviation $0.3$. The accurate samples are those that fall within $3\sigma$ of either Gaussian blob.

- **Product, Component B1:** Uniform samples along a circle of radius 1 centered at $(-0.5, 0)$.

- **Product, Component B2:** Identical to B1 but centered at $(0.5, 0)$.

- **Product, B1 $\wedge$ B2:** Two deterministic point masses at $(0, \pm\frac{\sqrt{3}}{2})$ (i.e., zero-variance Gaussians). Since the distribution has $0$ area of the support region, we define the accurate samples as those that fall within the distance of $0.1$ to either point mass.

- **Product, Component C1:** Uniform distribution in a left rectangle: $x \in [-1, -0.5], y \in [-1, 1]$.

- **Product, Component C2:** Uniform distribution in a right rectangle: $x \in [0.5, 1], y \in [-1, 1]$.

- **Product, C1 $\wedge$ C2:** Empty dataset (no samples are generated for the composed case).

- **Summation, Component A1:** Three Gaussians centered at $(-0.25, 0.5)$, $(-0.25, 0)$, and $(-0.25, -0.5)$, each with standard deviation $0.3$.

- **Summation, Component A2:** Three Gaussians centered at $(0.25, 0.5)$, $(0.25, 0)$, and $(0.25, -0.5)$, each with standard deviation $0.3$.

- **Summation, A1 $\vee$ A2:** Union of all six Gaussians from A1 and A2, each with standard deviation $0.3$. The accurate samples are those that fall within $3\sigma$ of any Gaussian component.

- **Summation, Component B1:** Uniform distribution in left rectangle: $x \in [-1, -0.5], y \in [-1, 1]$.

- **Summation, Component B2:** Uniform distribution in right rectangle: $x \in [0.5, 1], y \in [-1, 1]$.

- **Summation, B1 $\vee$ B2:** Union of samples from both rectangular regions. The accurate samples are those that fall on the support region of either rectangle.

- **Summation, Component C1:** Uniform samples along a unit circle centered at $(-0.5, 0)$.

- **Summation, Component C2:** Identical to C1 but centered at $(0.5, 0)$.

- **Summation, C1 $\vee$ C2:** Union of both circular distributions. Since the distribution as $0$ area of the support region, we define the accurate samples as those that fall within the distance of $[0.9, 1.1]$ to either center of the circle.

- **Negation, Component A1:** Uniform distribution over a square: $x, y \in [-1, 1]$.

- **Negation, Component A2:** Uniform distribution over the center square: $x, y \in [-0.5, 0.5]$.

- **Negation, A1 $\wedge\neg$ A2:** Union of four rectangular corner regions (i.e., all of A1 excluding A2). The accurate samples are those that fall on the support region of either rectangle corner.

- **Negation, Component B1:** Uniform distribution over square $[-1, 1] \times [-1, 1]$.

- **Negation, Component B2:** Uniform in a vertical strip: $x \in [-0.5, 0.5], y \in [-2, 2]$.

- **Negation, B1 $\wedge\neg$ B2:** Union of two vertical slabs: $x \in [-1, -0.5]$ and $x \in [0.5, 1], y \in [-1, 1]$. The accurate samples are those that fall on the support region of either slab.

- **Negation, Component C1:** Uniform in center square: $x, y \in [-0.5, 0.5]$.

- **Negation, Component C2:** Uniform in larger square: $x, y \in [-1, 1]$.

- **Negation, C1 $\wedge\neg$ C2:** Empty dataset (intended to represent the subtraction of C1 from C2, but not explicitly sampled).

### E.2. Notes on Why We Use Chamfer Distance as the Metric

The reader might ask why we choose Chamfer Distance as opposed to other metrics, e.g., Kullback–Leibler divergence, Wasserstein Distance, or others. We summarize the reasons as follows:

We chose Chamfer Distance because it (1) applies to uniform distributions and (2) is sensitive to out-of-distribution samples. KL is inapplicable for out-of-distribution samples with undefined density ratio in uniform distribution settings. Wasserstein Distance is more robust to outliers, which doesn't meet our requirement to sensitively capture unaligned samples.

# F. CLEVR Position Task

| Method | Combinations | | | | |
|---|---|---|---|---|---|
| | 1 | 2 | 3 | 4 | 5 |
| Composable Diffusion | **100.0** | **100.0** | 99.0 | 90.8 | 78.7 |
| EBM (U-HMC) | 82.0 | 66.0 | 34.0 | 24.0 | 11.0 |
| EBM (ULA) | 86.0 | 74.0 | 48.0 | 27.0 | 19.0 |
| Composable Diffusion + *Cached CompLift* | **100.0** | **100.0** | 99.7 | 91.6 | 84.3 |
| Composable Diffusion + *CompLift* | **100.0** | **100.0** | **99.9** | **97.5** | **90.3** |

*Table 5.* Quantitative accuracy results on CLEVR compositional generation (higher is better).

| Method | Combinations | | | | |
|---|---|---|---|---|---|
| | 1 | 2 | 3 | 4 | 5 |
| Composable Diffusion | 37.6 | 37.7 | 38.8 | **48.8** | **50.1** |
| EBM (U-HMC) | 158.9 | 139.7 | 111.3 | 101.4 | 84.4 |
| EBM (ULA) | 180.9 | 153.1 | 122.4 | 103.9 | 84.5 |
| Composable Diffusion + *Cached CompLift* | **36.8** | **36.2** | **36.4** | 50.2 | 57.3 |
| Composable Diffusion + *CompLift* | 38.2 | 36.4 | 40.1 | 53.2 | 56.2 |

*Table 6.* FID scores on CLEVR compositional generation (lower is better).

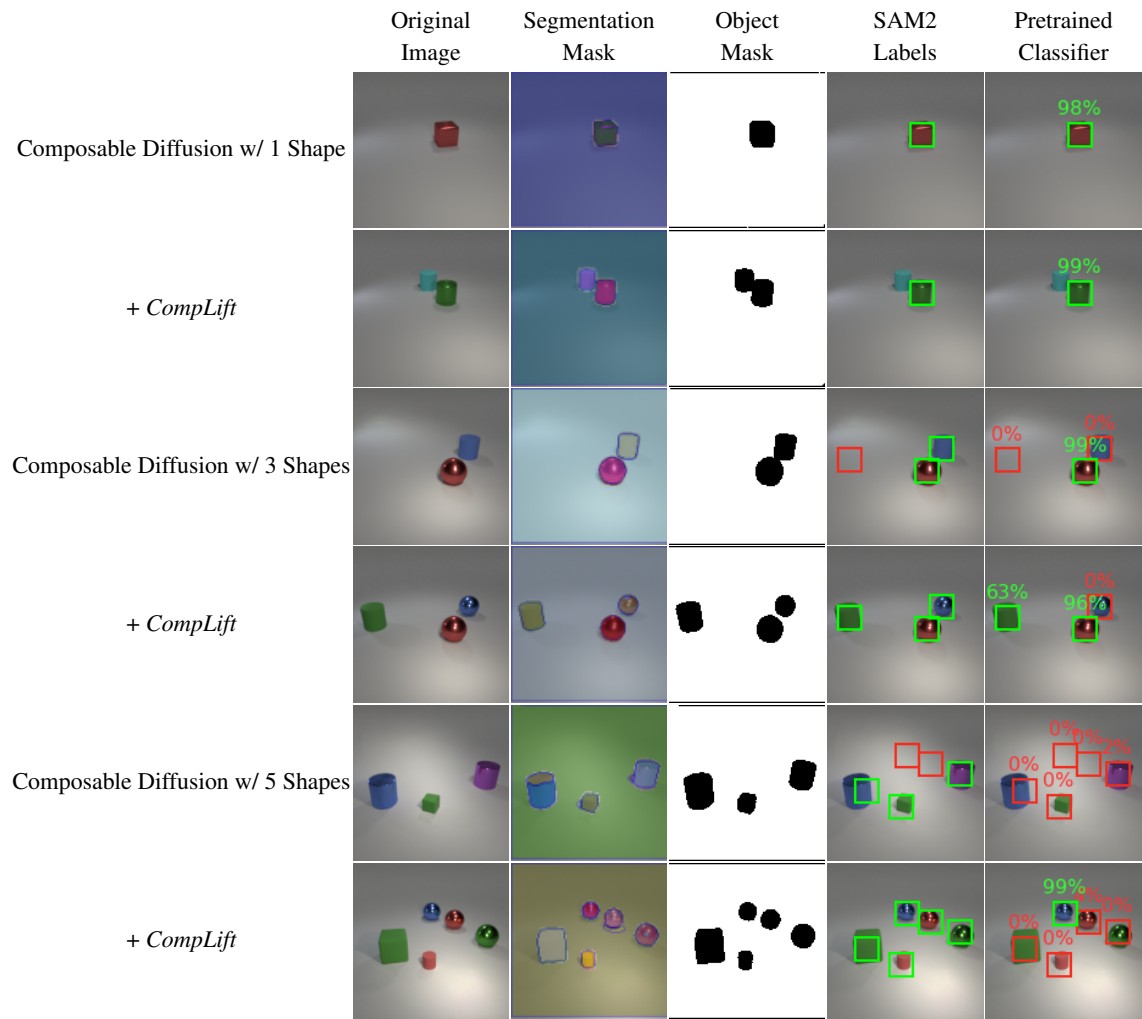

*Figure 18.* **More examples of samples on CLEVR Position dataset.** We find that the SAM verifier is more robust and generalizable compared to the pretrained classifier provided by the Composable Diffusion work (Liu et al., 2022). From left to right: the first column shows the original images, the second column shows the segmentation masks generated by SAM2, the third column shows the object masks (by excluding the pixels from the background mask), the fourth column shows the labels given by SAM2, and the fifth column shows the labels given by the pretrained classifier. The percentage numbers represent the confidence score predicted by the classifier.

We use Segment Anything Model 2 (SAM2) (Ravi et al., 2024) as a generalizable verifier, to check whether an object is in a specified position. We first extract the background mask, which is the mask with the largest area using the automatic mask generator of SAM2 (SAM2AutomaticMaskGenerator), then label a condition as satisfied if the targeted position is not in the background mask. We find this new verifier is more robust than the pretrained classifier in Liu et al. (2022). As shown in Figure 18, the pretrained classifier often fails to generalize when more shapes are required to be composed, while the SAM2 verifier can effectively detect the samples due to the fact that it is pretrained with massive segmentation dataset.

For the MCMC implementation, we use the same PyTorch samplers shared by Du et al. (Du et al., 2023) [1]. However, we find it hard to reproduce the result of the MCMC method on the CLEVR Position Task. The generated objects in the synthesized images are often in strange shapes that are dissimilar from the trained data. We suspect that the MCMC method may require more careful tuning of hyperparameters, as not all hyperparameters are mentioned as the optimal ones in the codebase. *Therefore, we remind the readers that the results of the MCMC method on the CLEVR Position Task in this section should be interpreted with caution - they mainly serve as references and do not represent the optimal results.*

Nevertheless, we are still interested in how the methods perform using the pretrained classifier from Liu et al. (Liu et al., 2022). In Table 7, we find that the Composable Diffusion model with *CompLift* consistently outperforms the baseline methods across all combinations, and achieves a nonzero accuracy in the most challenging 5 object composition setting.

| Method | Combinations | | | | |
|---|---|---|---|---|---|
| | 1 | 2 | 3 | 4 | 5 |
| Composable Diffusion | 52.7 | 20.5 | 3.1 | 0.3 | 0.0 |
| EBM (U-HMC) | 1.0 | 0.0 | 0.0 | 0.0 | 0.0 |
| EBM (ULA) | 6.0 | 0.0 | 0.0 | 0.0 | 0.0 |
| Composable Diffusion + *Cached CompLift* | 55.7 | **25.3** | **6.7** | 0.4 | 0.0 |
| Composable Diffusion + *CompLift* | **56.2** | 25.3 | 5.7 | **0.7** | **0.1** |

*Table 7.* Quantitative accuracy results on CLEVR using the pretrained classifier from Liu et al. (Liu et al., 2022).

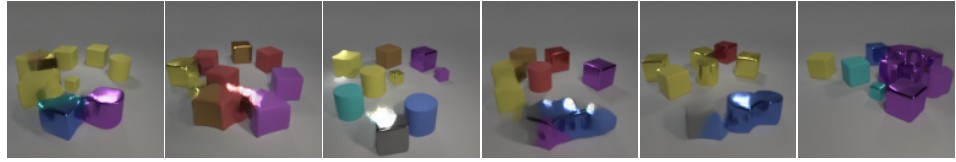

*Figure 19.* **Example of failed samples on CLEVR Position dataset using MCMC.** The generated objects in the synthesized images are often in strange shapes which is dissimilar from the trained data. We suspect that the MCMC method may require a more careful tuning of hyperparameters. As a result, we remind the readers that the results of the MCMC method on the CLEVR Position Task should be interpreted with caution.

[1] https://github.com/yilundu/reduce_reuse_recycle/blob/main/anneal_samplers.py

# G. Text-to-Image Generation

## G.1. More Examples on Correlation Between CLIP and *CompLift*

In Figures 10 and 20, we provide examples of the correlation between CLIP and *CompLift* for text-to-image generation. The figures show the generated images for 3 prompts involving *a black car and a white clock*, *a turtle with a bow* and *a frog and a mouse*. For these prompts, we find that the diffusion models would typically ignore the clock, bow, and mouse components, leading to low CLIP scores for many generated images. Meanwhile, we observe a strong correlation between the CLIP and *CompLift* scores for these prompts, indicating the effectiveness of *CompLift* for complex prompts. For the images with low CLIP scores, they typically have lower number of activated pixels using *CompLift* for these ignored objects.

One note is that for the *a turtle with a bow* prompt, we observe that most of the images would fail to generate the bow component. This explains why many data samples have low CLIP scores on the left part of Figure 20. However, the *CompLift* algorithm can effectively capture the bow component, and reject those samples with low CLIP score.

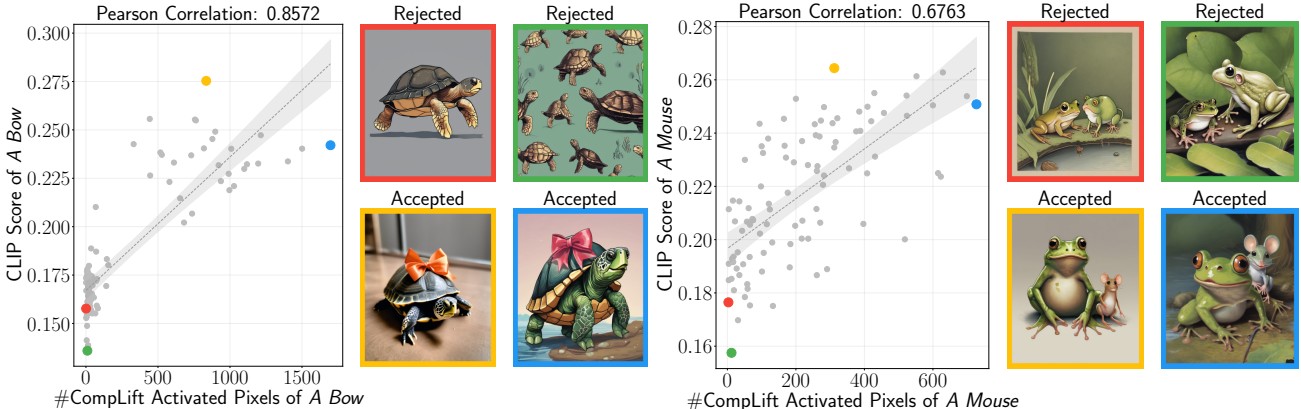

*Figure 20.* **More examples of correlation between CLIP and *CompLift*.** Prompt to generate left images: a turtle with a bow. Prompt to generate right images: a frog and a mouse. Text in blue indicates the object components to compose.

## G.2. Ablation Study: Improvement on Attend&Excite Generator

We conducted a new experiment using Attent-and-Excite (Chefer et al., 2023), focusing on the additional improvement achieved by incorporating CompLift. We observed consistent performance gains with both SD 1.4 and SD 2.1. Note that SD XL is not included due to the lack of support in the original Attent-and-Excite code.

| Method | Animals | | Object&Animal | | Objects | |
|---|---|---|---|---|---|---|
| | CLIP ↑ | IR ↑ | CLIP ↑ | IR ↑ | CLIP ↑ | IR ↑ |
| A&E (SD 1.4) | 0.330 | 0.831 | 0.357 | 1.339 | 0.357 | 0.815 |
| A&E (SD 1.4) + *Cached CompLift* | 0.338 | 1.156 | 0.361 | 1.469 | 0.362 | 0.934 |
| A&E (SD 1.4) + *CompLift (T=200)* | 0.337 | **1.160** | 0.361 | 1.458 | 0.361 | 0.990 |
| A&E (SD 2.1) | 0.342 | 1.225 | 0.360 | 1.471 | 0.366 | 1.219 |
| A&E (SD 2.1) + *Cached CompLift* | 0.344 | 1.298 | 0.364 | 1.488 | **0.371** | 1.245 |
| A&E (SD 2.1) + *CompLift (T=200)* | **0.346** | **1.337** | **0.365** | **1.516** | 0.370 | **1.246** |

*Table 8.* CLIP and IR scores across categories for Attend-and-Excite (A&E) with CompLift variants.

## G.3. Ablation Study: Advantage of Compositional Criteria

One concern is that the compositional acceptance / rejection task can also be framed using 1 single criterion that works directly on the whole prompt, as addressed by CAS (Hong et al., 2024). To test how CAS-like variant performs for prompts containing multiple objects, we have conducted a new ablation study.

Here, CAS-like variant means that we use the single criteria $\log p(z \mid c_{\text{compose}}) - \log p(z \mid \varnothing)$ as the latent lift score, which replaces the composed criteria from multiple individual lift scores in CompLift. Note that this is a controlled experiment to

check the advantage of compositional criteria, thus, we keep the same estimation method using ELBO.

We provide Table 9 as the result. We observe only modest improvement when using the CAS-like variant. We hypothesize that CAS-like variant might face a similar problem as the original Diffusion Model for multi-object prompts - the attention to the missing object is relatively weak in the attention layers. More similar discussions can be found in previous works such as Attend-and-Excite (Chefer et al., 2023), where diffusion model $\epsilon_\theta(x, c_{\text{compose}})$ sometimes ignores some condition $c_i$ in $c_{\text{compose}}$.

| Method | Animals | | Object&Animal | | Objects | |
|---|---|---|---|---|---|---|
| | CLIP ↑ | IR ↑ | CLIP ↑ | IR ↑ | CLIP ↑ | IR ↑ |
| SD 1.4 | 0.310 | -0.191 | 0.343 | 0.432 | 0.333 | -0.684 |
| SD 1.4 + *CAS Variant* | 0.312 | -0.153 | 0.348 | 0.708 | 0.337 | -0.373 |
| SD 1.4 + *CompLift (T=200)* | **0.322** | **0.293** | **0.358** | **1.093** | **0.347** | **-0.050** |
| SD 2.1 | 0.330 | 0.532 | 0.354 | 0.924 | 0.342 | -0.112 |
| SD 2.1 + *CAS Variant* | 0.333 | 0.626 | 0.355 | 1.080 | 0.347 | 0.144 |
| SD 2.1 + *CompLift (T=200)* | **0.340** | **0.975** | **0.362** | **1.283** | **0.355** | **0.489** |
| SD XL | 0.338 | 1.025 | 0.363 | 1.621 | 0.359 | 0.662 |
| SD XL + *CAS Variant* | 0.338 | 1.064 | 0.363 | 1.628 | 0.362 | 0.702 |
| SD XL + *CompLift (T=200)* | **0.342** | **1.216** | **0.364** | **1.706** | **0.367** | **0.890** |

*Table 9.* Performance comparison (CLIP and IR) across categories for SD backbones, with CAS Variant and CompLift.

### G.4. More Examples on Accepted and Rejected Images

In Figures 21 and 22, we provide additional examples of accepted and rejected images using SDXL + *CompLift*. We observe that the *CompLift* algorithm can effectively reject samples that fail to capture the object components specified in the prompt in general. In addition, we find that the current *CompLift* criterion is relatively weak in determining the color attribute, leading to some accepted images with incorrect colors (e.g., *a green backpack and a purple bench*). This suggests that future work could explore more sophisticated criteria to further improve the quality of generated images.

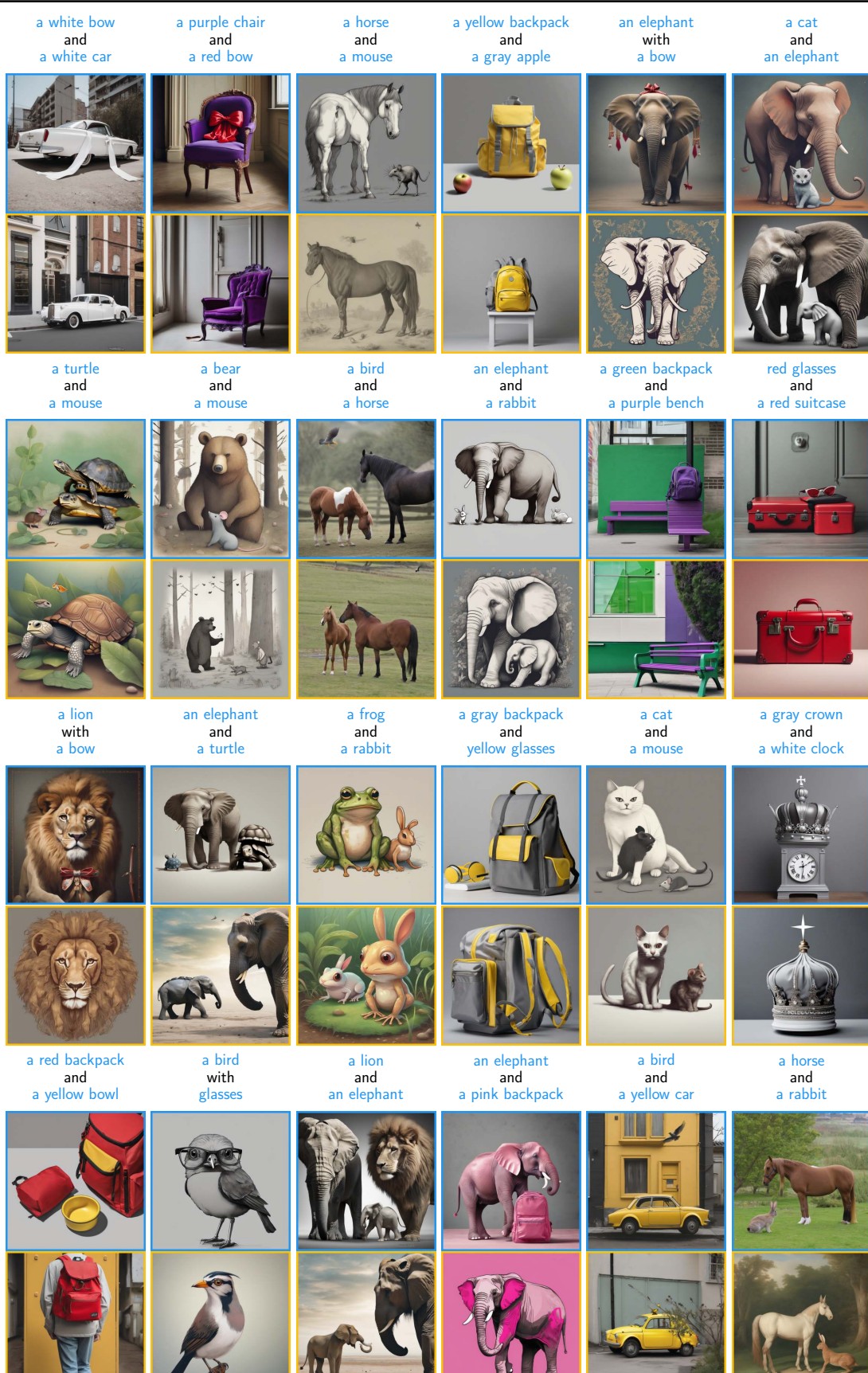

*Figure 21.* More examples of accepted and rejected images using SDXL + *CompLift* for text-to-image generation.

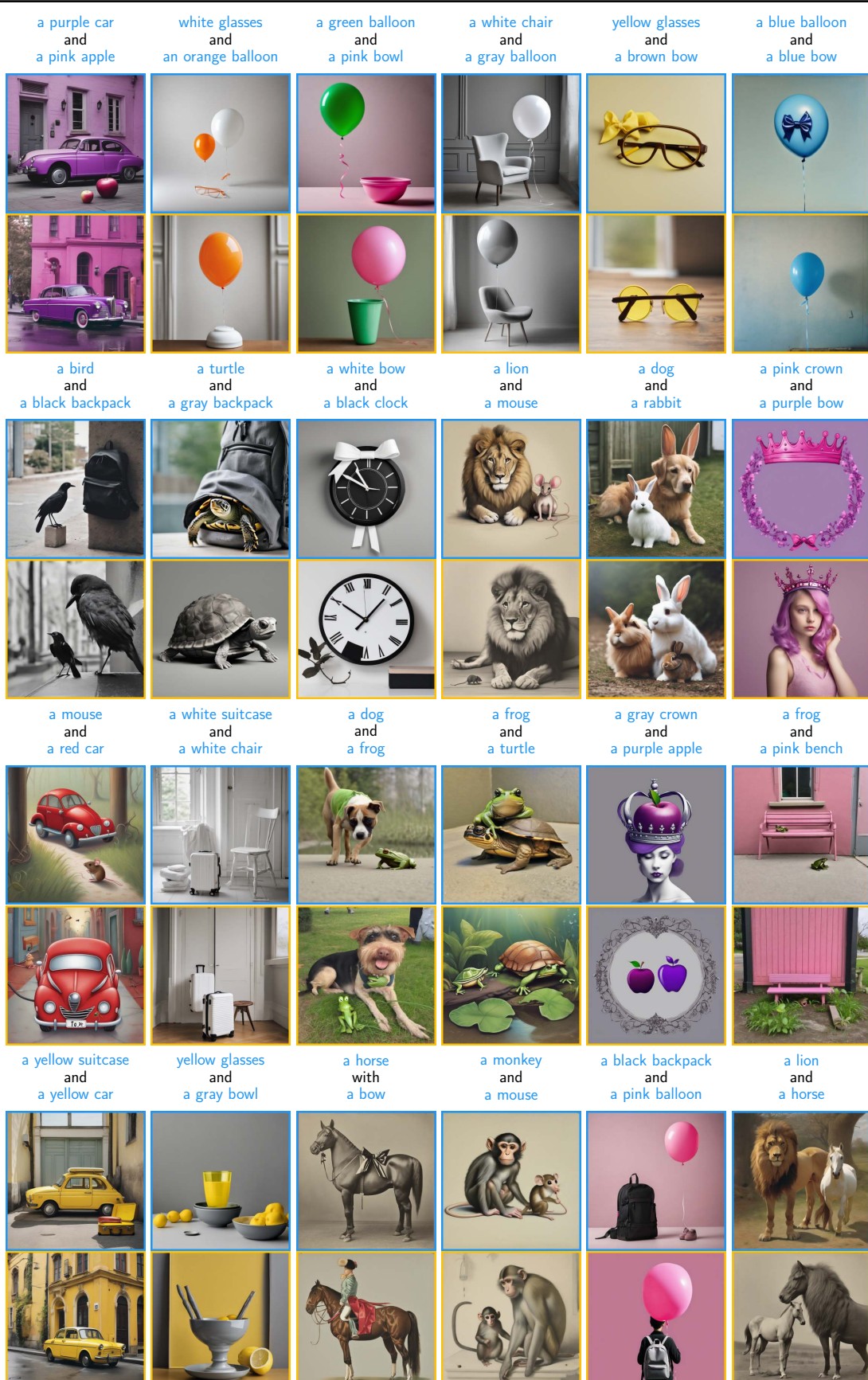

*Figure 22.* More examples of accepted and rejected images using SDXL + *CompLift* for text-to-image generation.

# H. Q&A during Rebuttal

During the rebuttal, we have received some questions from the reviewers. We appreciate the valuable feedback from anonymous reviewers, and we incorperate our answers in the rebuttal as follows:

**Q: Though CompLift shows significant improvement over Composable Stable Diffusion on synthetic datasets, the improvement on real-world text-to-image generative model is trivial as shown in Table 3.**

A: The seemingly trivial CLIP improvement is due to the low magnitude of CLIP scores. Here, we provide another perspective to interpret the numbers. We compare the CompLift selector to the perfect best-of-n selector, which has direct access to the metric function. The percentage gain is calculated as (CompLift metric - baseline metric) / (perfect selector metric - baseline metric). On average, the gains are 40% for vanilla CompLift and 30% for cached CompLift.

| Method | Animals | | Object&Animal | | Objects | |
|---|---|---|---|---|---|---|
| | CLIP gain% ↑ | IR gain% ↑ | CLIP gain% ↑ | IR gain% ↑ | CLIP gain% ↑ | IR gain% ↑ |
| SD 1.4 + *Cached CompLift* | 31.58 | 30.58 | 42.62 | 62.74 | 37.04 | 54.99 |
| SD 1.4 + *CompLift (T=200)* | **42.11** | **46.40** | **49.18** | **74.32** | **47.14** | **63.05** |
| SD 2.1 + *Cached CompLift* | 35.71 | 44.13 | 28.34 | 55.07 | 37.97 | 57.82 |
| SD 2.1 + *CompLift (T=200)* | **39.68** | **56.18** | **32.39** | **60.28** | **41.14** | **74.73** |
| SD XL + *Cached CompLift* | 16.95 | **55.88** | **26.13** | 46.15 | 24.49 | **46.06** |
| SD XL + *CompLift (T=200)* | **22.60** | 48.74 | **26.13** | **59.44** | **32.65** | 44.88 |

*Table 10.* CLIP and IR gain (%) across categories with CompLift variants.

**Q: Would it be feasible to evaluate with the TIFA score as in Karthik et al (Karthik et al., 2023; Hu et al., 2023)?**

A: New TIFA experiments show improvements across categories:

| Method | Animals TIFA ↑ | Object&Animal TIFA ↑ | Objects TIFA ↑ |
|---|---|---|---|
| Stable Diffusion 1.4 | 0.692 | 0.822 | 0.629 |
| SD 1.4 + *Cached CompLift* | 0.750 | 0.886 | **0.685** |
| SD 1.4 + *CompLift (T=200)* | **0.794** | **0.902** | 0.682 |
| Stable Diffusion 2.1 | 0.833 | 0.873 | 0.668 |
| SD 2.1 + *Cached CompLift* | 0.905 | 0.911 | **0.731** |
| SD 2.1 + *CompLift (T=200)* | **0.927** | **0.912** | 0.726 |
| Stable Diffusion XL | 0.913 | 0.964 | 0.755 |
| SD XL + *Cached CompLift* | **0.949** | 0.972 | **0.790** |
| SD XL + *CompLift (T=200)* | 0.946 | **0.974** | 0.782 |

*Table 11.* TIFA results across categories. CompLift variants show consistent improvements over baselines.

**Q: For assessing alignment via CLIP do you use the entire prompt (including all conditions) or assess the CLIP alignment on each condition individually? I believe it's the former but do you think they latter might work better? (do you have any evidence on this?)**

A: We used entire prompt. We add new experiment with minCLIP (minimum CLIP score across subjects) in Table 12. Similar performance gains were observed, indicating CompLift primarily improves the weaker condition (typically missing object).

| Method | Animals minCLIP ↑ | Object&Animal minCLIP ↑ | Objects minCLIP ↑ |
|---|---|---|---|
| Stable Diffusion 1.4 | 0.218 | 0.248 | 0.237 |
| SD 1.4 + *Cached CompLift* | 0.225 | 0.260 | 0.249 |
| SD 1.4 + *CompLift (T=200)* | **0.228** | **0.263** | **0.252** |
| Stable Diffusion 2.1 | 0.237 | 0.258 | 0.247 |
| SD 2.1 + *Cached CompLift* | 0.248 | **0.265** | 0.260 |
| SD 2.1 + *CompLift (T=200)* | **0.249** | **0.265** | **0.261** |
| Stable Diffusion XL | 0.243 | 0.269 | 0.264 |
| SD XL + *Cached CompLift* | 0.248 | **0.271** | 0.269 |
| SD XL + *CompLift (T=200)* | **0.250** | **0.271** | **0.271** |

*Table 12.* minCLIP scores across categories, showing improvements with CompLift variants.

**Q: Do you have any way to assess whether CLIP vs ImageReward is a more appropriate metric, and how well each of them actually checks whether all the conditions are present in the composition (e.g. for AND).**

A: We manually labeled 100 samples from Figure 10 for presence of both black car and white clock. With 62 positive and 38 negative samples, we calculated metric performance in Table 13. CLIP and ImageReward perform similarly, both better than TIFA. CLIP slightly preferred due to imbalanced data.

| Metrics | CLIP | ImageReward | TIFA |
|---|---|---|---|
| ROC AUC | 0.949 | 0.955 | 0.857 |
| PR AUC | 0.972 | 0.968 | 0.901 |

*Table 13.* ROC AUC and PR AUC for CLIP, ImageReward, and TIFA on the correspondence of the presence of both black car and white clock. Experiments were conducted on 100 sampled images from Figure 10.

