# OpenReview forum: "Improving Compositional Generation with Diffusion Models Using Lift Scores"
_ICML.cc/2025/Conference — ICML 2025 poster_

### Official Review · Reviewer_xPQc · 2025-03-04

**Overall Recommendation:** 3

**Summary:**

This paper aims to improve compositional generation at inference time via rejection sampling using Lift scores on each condition to be composed.

**Update after rebuttal**
I appreciate the additional data (which I found convincing) and clarifications provided by the authors during the rebuttal. I was previously unaware of the prior work CAS mentioned by reviewer kyUu and agree that it may impact the novelty, but I also found the authors' clarifications about their focus specifically on compositions and systematic evaluation fairly convincing. Overall, I will keep my score at 3.

**Claims And Evidence:**

Yes

**Essential References Not Discussed:**

N/A

**Experimental Designs Or Analyses:**

Please see Methods And Evaluation Criteria

**Methods And Evaluation Criteria:**

Quantitative metrics are important for this study. I feel that this was done thoroughly and mostly reasonable choices were made, although I have a few questions.

2D synthetic: (Q) Why did you choose Chamfer Distance (as opposed to other metrics e.g. KL, Wasserstein etc.)?

CLEVR: SAM2 and Liu pretrained classifier both make sense, it is nice that they were both tried and compared.

SDXL: CLIP and ImageReward seem reasonable here.
(Q) Could Segment Anything potentially be used in this context? (why not?).
(Q) Would it be feasible to evaluate with the TIFA score as in Karthik et al (https://arxiv.org/pdf/2305.13308)?
(Q) (L407) For assessing alignment via CLIP do you use the entire prompt (including all conditions) or assess the CLIP alignment on each condition individually? I believe it’s the former but do you think they latter might work better? (do you have any evidence on this?)
(Q) Do you have any way to assess whether CLIP vs ImageReward is a more appropriate metric, and how well each of them actually checks whether *all* the conditions are present in the composition (e.g. for AND).

**Other Comments Or Suggestions:**

I would appreciate a clearer discussion of the final probability you are actually optimizing for (or energy you are minimizing) after your rejection-sampling procedure, if there is anything you can say theoretically about it. I did not find the variance-reduction interpretation of replacing epsilon by eps_theta(x, c_compose) very clear. I also wonder if there are any connections with CFG that you know of?

**Other Strengths And Weaknesses:**

Strengths: I appreciate the focus on the missing objects problem. I find the samples and metrics fairly convincing (though please see Questions).

Weaknesses: A lot of the decisions feel a bit ad-hoc (e.g. how many activated pixels “count” e.g. tau = 250 on L245 and the choice to replace epsilon by eps_theta(x, c_compose) in Fig 6) although I don’t consider this a deal-breaker for a methods paper.

**Questions For Authors:**

Please see the questions in Methods And Evaluation Criteria.

A few others:
Are Figure 4, 7, samples cherrypicked? If so I feel this should be acknowledged in the caption, and possibly some not-so-great examples included in appendix.

Answers to these questions would help me find the samples and metrics more convincing.

**Relation To Broader Scientific Literature:**

This approach builds on various existing approaches in diffusion, composition, rejection sampling, Lift scores, etc. I don't feel that it represents a big conceptual leap beyond these existing methods, however I think it cites prior work appropriately and focuses mainly on getting a conceptually simple idea to work well empirically with suitable validation.

**Theoretical Claims:**

N/A

---

> ### Author Rebuttal · Authors · 2025-03-30
>
> Thank you for your valuable comments. We address your concerns as follows:
>
> > Chamfer Distance
>
> We chose Chamfer Distance because it (1) applies to uniform distributions and (2) is sensitive to out-of-distribution samples. KL is inapplicable for out-of-distribution samples with undefined density ratio in uniform distribution settings. Wasserstein Distance is more robust to outliers, which doesn't meet our requirement to sensitively capture unaligned samples.
>
> >  Segment Anything for t2i
>
> Vanilla Segment Anything lacks text prompt support. Grounded Segment Anything [1] would be a good candidate. We used CLIP/ImageReward following standards in previous works [2,3], but we will add a discussion in Conclusion.
>
> > TIFA score [2, 4]
>
> New TIFA experiments show improvements across categories:
>
> | Method | Animals | Object&Animal | Objects |
> | --- | :---: | :---: | :---: |
> |  | TIFA ↑ | TIFA ↑ | TIFA ↑ |
> | Stable Diffusion 1.4 | 0.692 | 0.822 | 0.629 |
> | SD 1.4 + *Cached CompLift* | 0.750 | 0.886 | **0.685** |
> | SD 1.4 + *CompLift* | **0.794** | **0.902** | 0.682 |
> | | | | |
> | Stable Diffusion 2.1 | 0.833 | 0.873 | 0.668 |
> | SD 2.1 + *Cached CompLift* | 0.905 | 0.911 | **0.731** |
> | SD 2.1 + *CompLift* | **0.927** | **0.912** | 0.726 |
> | | | | |
> | Stable Diffusion XL | 0.913 | 0.964 | 0.755 |
> | SD XL + *Cached CompLift* | **0.949** | 0.972 | **0.790** |
> | SD XL + *CompLift* | 0.946 | **0.974** | 0.782 |
>
> > entire prompt vs individual condition
>
> We used entire prompt. We add new experiment with minCLIP (minimum CLIP score across subjects):
>
> | Method | Animals | Object&Animal | Objects |
> | --- | :---: | :---: | :---: |
> |  | minCLIP ↑ | minCLIP ↑ | minCLIP ↑ |
> | Stable Diffusion 1.4 | 0.218 | 0.248 | 0.237 |
> | SD 1.4 + *Cached CompLift* | 0.225 | 0.260 | 0.249 |
> | SD 1.4 + *CompLift* | **0.228** | **0.263** | **0.252** |
> |  |  |  |  |
> | Stable Diffusion 2.1 | 0.237 | 0.258 | 0.247 |
> | SD 2.1 + *Cached CompLift* | 0.248 | **0.265** | 0.260 |
> | SD 2.1 + *CompLift* | **0.249** | **0.265** | **0.261** |
> |  | |  |  |
> | Stable Diffusion XL | 0.243 | 0.269 | 0.264 |
> | SD XL + *Cached CompLift* | 0.248 | **0.271** | 0.269 |
> | SD XL + *CompLift* | **0.250** | **0.271** | **0.271** |
>
> Similar performance gains were observed, indicating CompLift primarily improves the weaker condition (typically missing object).
>
> > CLIP vs ImageReward
>
> We manually labeled 100 samples from Fig. 10 for presence of both black car and white clock. With 62 positive and 38 negative samples, we calculated metric performance:
>
> | Metrics | CLIP | ImageReward | TIFA |
> | --- | :---: | :---: | :---: |
> | ROC AUC | 0.949 | 0.955 | 0.857 |
> | PR AUC | 0.972 | 0.968 | 0.901 |
>
> CLIP and ImageReward perform similarly, both better than TIFA. CLIP slightly preferred due to imbalanced data.
>
> > decisions feel a bit ad-hoc
>
> We choose $\tau=250$ as the median activated pixel count among all images. Tests at 25th and 75th percentiles showed the median works best - lower $\tau$ reduces accuracy due to estimation variance, while higher $\tau$ increase rejection rates.
>
> Regarding $\epsilon_\theta(x_t, c_{compose})$: this design is from empirical observations. Intuitively, if object $c$ exists in image $x$, then for most noisy images $x_t$, $\epsilon_\theta(x_t, c)$ should be closer to $\epsilon_\theta(x_t, c_{compose})$ than the unconditional $\epsilon_\theta(x_t, \varnothing)$ in the corresponding pixels. We'll add this explanation to the paper.
>
> > connections with CFG
>
> Our paper uses the constrained distribution:
> \begin{equation}
>  x_0 \sim p_{\text{generator}}(x_0), \quad \text{s.t. } \log p(x_0 \mid c_i) - \log p(x_0) > 0, \quad \forall\, c_i.
> \end{equation}
>
> Now we show that [5,6] tries to satisfy the constraint using soft regularization. Using Lagrangian relaxation and multipliers $\lambda_i \geq 0$, the objective can be transformed into:
> \begin{equation}
> \mathcal{L}(x_0, \lambda) = \log p_{\text{generator}}(x_0) + \sum_{c_i} \lambda_i \Bigl( \log p(x_0 \mid c_i) - \log p(x_0) \Bigr), \quad \lambda_i \geq 0.
> \end{equation}
>
> Since
> $\nabla_{x_t}\log p_\theta(x_0) \propto \epsilon_\theta(x_t, t)$, and [5, 6] assume an unconditional generator,
> the derivative matches Equation 11 in [6]:
> \begin{equation}
> \nabla_{x_t}\mathcal{L}(x_0, \lambda) \propto \epsilon_\theta(x_t, t) + \sum_{c_i} \lambda_i \Bigl( \epsilon_\theta(x_t, t \mid c_i) - \epsilon_\theta(x_t, t) \Bigr), \quad \lambda_i \geq 0.
> \end{equation}
>
> CFG [5,6] uses fixed $\lambda_i=w$, not guaranteeing constraint satisfaction.
>
> > Are Figure 4, 7, samples cherrypicked?
>
> Yes: Figure 4 shows the ones with most-improved CLIP scores. Figure 7 uses random prompts with clear pixel separation and aesthetic quality. We will add more samples to Appendix and captions to make the selection clear.
>
> [1] https://arxiv.org/abs/2401.14159
>
> [2] https://arxiv.org/abs/2305.13308
>
> [3] https://arxiv.org/abs/2301.13826
>
> [4] https://arxiv.org/abs/2303.11897
>
> [5] https://arxiv.org/abs/2207.12598
>
> [6] https://arxiv.org/abs/2206.01714

---

### Official Review · Reviewer_kdTw · 2025-03-10

**Overall Recommendation:** 3

**Summary:**

- The paper introduces a novel criterion CompLift for rejecting samples of conditional diffusion models based on lift scores.
- For compositional generation, i.e., cases in which the condition for sampling, e.g, a text prompt can be described as a composition of conditions (like desired individual objects in the image), CompLift intuitively evaluates whether final samples are more likely given each individual condition than without it and therefore whether the conditions have been properly considered throughout the generation process.
- Formally, this criterion can be described in terms of lift scores, an existing concept in data mining, for which the authors introduce an approximation using the same conditional diffusion model as for sampling.
- In an exploration of the design space, the authors evaluate the effect of noise and timestep sampling for the approximation and propose a more efficient algorithm that caches intermediate results from the generation process for later evaluation of the rejection criterion.
- An evaluation on synthetic data, a toy image dataset, as well as text-to-image generation shows improved alignment with the conditions for compositional generation.

**Claims And Evidence:**

Most claims in the submission are supported by clear and convincing evidence except for:

- The paper claims "significantly improve[d] compositional generation" (lines 24 ff., left column) while the quantitative results on the CLEVR position dataset mainly show accuracy improvements with 4 and 5 constraints, for which the FID however is worse than the Composable Diffusion baseline as also mentioned by the authors (lines 424 ff., left column)
  - If the rejection of samples reduces sample diversity as hypothesized by the authors, the kind of improvements for compositional generation (condition / prompt alignment) should be specified to avoid misunderstandings.
- The paper compares quantitatively on the 2D synthetic dataset against additional baselines (EBM [1]), but limits qualitative comparisons to the Composable Diffusion baseline only.
- The main paper compares only on the 2D synthetic dataset against baselines (EBM [1]), but misses to do so on the CLEVR and text-to-image setups. The appendix provides comparisons on CLEVR.

[1] Reduce, Reuse, Recycle: Compositional Generation with Energy-Based Diffusion Models and MCMC. ICML 2023

**Essential References Not Discussed:**

I am not aware of any essential missing references.

**Experimental Designs Or Analyses:**

All experimental designs and analyses seem to be valid.

**Methods And Evaluation Criteria:**

The proposed methods and evaluation criteria make sense except for:
- For the text-to-image compositional task, the number of trials is different for the vanilla version and the cached version. While I understand that the number of trials for the cached version is equal to the number of sampling steps, a fair comparison of both versions in terms of number of trials would be interesting to see for this task in order to evaluate the effect of caching.
- While providing results on the 2D synthetic dataset makes sense, there is a severe lack of clarity regarding this benchmark:
  - Figure 1 showing results on this dataset is already on page 2 but first referenced on page 7. The caption of it also does not describe the experimental setup. I found this figure to be unclear if the dataset is not introduced yet.
  - As a result of that, the Compose function (lines 119 ff., right column) together with table 1 lack intuition. Having the synthetic 2D dataset described earlier for an example of different compositions of conditions (or possibly a text-to-image example in the introduction using different compose functions) would be helpful.
  - The introduction of the dataset in section 6 and metrics after the design space exploration / ablation (section 4) with quantitative resutls on this benchmark and figures 2 and 3 also raise questions about the dataset and how the accuracy is measured while reading the paper.
  - The description of the 2D synthetic dataset at the beginning of section 6.1 is lacking information about how the dataset is generated.

**Other Comments Or Suggestions:**

- In line 194 f., right column, you reference section 4.3. in section 4.3. itself.
- In line 317, left column, the $z$ should be a $z_t$, if I am not mistaken.

**Other Strengths And Weaknesses:**

Strengths:
- The paper is mostly well-written and easy to follow. Abstract, introduction, and related work provide a good motivation and introduction into the topic of compositional generation.
- The method section includes formal derivations of the lift score approximation as well as the intuition behind the equations, which I found very helpful.
- The qualitative results are convincing:
  - Once the 2D synthetic dataset and task is understood, the qualitative results illustrate the effect of CompLift and the different Compose functions well.
  - The pixel-wise scores for text-to-image generation clearly decompose the image according to the individual conditions.
- The quantitative results show consistent improvements over the Composable Diffusion baseline.

Weaknesses:
- As already indicated in review section "Methods And Evaluation Criteria", the structure of the paper w.r.t. the 2D synthetic dataset, figure 1, the different compose functions with table 1, and section 4 consisting of ablations before the dataset description is suboptimal and results in a lack of clarity.
  - And the dataset description itself also lacks information about how it was generated.
- More lack of clarity:
  - In lines 154 ff., right column, the explanation for the small performance loss using noise sharing strategies in negation tasks is unclear to me.
  - I find the description of the cached CompLift version in Section 4.3. and Algorithm 2 difficult to understand, even though the idea itself is quite simple and intuitive.
- The paper never introduces the abbreviations for the baselines from EBM and also not EBM itself.

**Questions For Authors:**

I do not have any particular questions to the authors.

**Relation To Broader Scientific Literature:**

Given complex conditions such as text prompts, diffusion models are known to hallucinate samples not following the correct conditional distribution. If conditions can be decomposed into smaller conditions, prior work like Composable Diffusion [1] has shown that this compositionality can be effectively leveraged to generate samples with better alignment to complex prompts. This paper proposes a method orthogonal to prior work by introducing a rejection / resampling criterion to ensure that the final sample is positively correlated with the conditions.

**Theoretical Claims:**

There are no theoretical claims that require proofs.

---

> ### Author Rebuttal · Authors · 2025-03-30
>
> Thank you for your valuable feedback. We address your concerns below.
>
> > Overstatement of minimal
>
> We will modify the abstract to make the contribution accurate as "significantly improved the condition alignment for compositional generation".
>
> > Limited comparisons: no T=50 for vanilla CompLift; missing EBM baselines on CLEVR/text-to-image
>
> We have conducted additional experiments to address both concerns:
>
> 1) We compared against EBM+ULA [1] on text-to-image generation (ULA is the default text-to-image sampler in their repo).
>
> 2) We ran vanilla CompLift with T=50 to enable fair comparison.
>
> The consolidated results are in the table below. CompLift outperforms EBM+ULA across all model variants, and the cached version performs similarly to vanilla CompLift with the same T=50, with CLIP scores very close while ImageReward scores are slightly lower for the cached version.
>
> We wish to clarify that our main focus is demonstrating vertical improvement - how CompLift boosts the base method's performance. The horizontal comparison to other baselines serves as supportive evidence that this boost helps achieve state-of-the-art performance.
>
> | Method | Animals |  | Object&Animal |  | Objects |  |
> |--------|---------|---------|---------|---------|---------|---------|
> |        | CLIP ↑ | IR ↑ | CLIP ↑ | IR ↑ | CLIP ↑ | IR ↑ |
> | Stable Diffusion 1.4 | 0.310 | -0.191 | 0.343 | 0.432 | 0.333 | -0.684 |
> | SD 1.4 + EBM (ULA) | 0.311 | 0.026 | 0.342 | 0.387 | 0.344 | -0.380 |
> | SD 1.4 + *Cached CompLift* | 0.319 | 0.128 | 0.356 | 0.990 | 0.344 | -0.131 |
> | SD 1.4 + *CompLift* (T=50) | 0.320 | 0.241 | 0.355 | 0.987 | 0.344 | -0.154 |
> | SD 1.4 + *CompLift* (T=200) | **0.322** | **0.293** | **0.358** | **1.093** | **0.347** | **-0.050** |
> ||||||||
> | Stable Diffusion 2.1 | 0.330 | 0.532 | 0.354 | 0.924 | 0.342 | -0.112 |
> | SD 2.1 + EBM (ULA) | 0.330 | 0.829 | 0.357 | 0.981 | 0.348 | 0.218 |
> | SD 2.1 + *Cached CompLift* | 0.339 | 0.880 | 0.361 | 1.252 | 0.354 | 0.353 |
> | SD 2.1 + *CompLift* (T=50) | **0.340** | **0.992** | 0.361 | 1.263 | 0.354 | 0.454 |
> | SD 2.1 + *CompLift* (T=200) | **0.340** | 0.975 | **0.362** | **1.283** | **0.355** | **0.489** |
> ||||||||
> | Stable Diffusion XL | 0.338 | 1.025 | 0.363 | 1.621 | 0.359 | 0.662 |
> | SD XL + EBM (ULA) | 0.335 | 0.913 | 0.362 | 1.676 | 0.361 | 0.872 |
> | SD XL + *Cached CompLift* | 0.341 | **1.244** | **0.364** | 1.687 | 0.365 | **0.896** |
> | SD XL + *CompLift* (T=50) | **0.342** | 1.222 | **0.364** | 1.700 | 0.365 | 0.842 |
> | SD XL + *CompLift* (T=200) | **0.342** | 1.216 | **0.364** | **1.706** | **0.367** | 0.890 |
>
>
> > Structure issues
>
> Thank you for the constructive feedback. We will try our best to make the concept more clear. In particular, we will make the following modifications:
> 1. Self-inclusive caption - summarize the experiment setup in the caption of Figure 1, including the component distribution, the algebra, the data generation, and the training.
> 2. Early introduction of 2D dataset - briefly mention the 2D synthetic dataset in Section 3, including the description about the data generation, the component distribution, the algebra, and the accuracy metric. We will add a sentence to refer reader to Section 6 and Appendix D for more details.
> 3. Intuitive explanation - add several more explanation of the Compose function, using the examples in the 2D dataset.
>
> > 2D dataset missing generation details
>
> We will add more text to Section 6.1 about the dataset generation. In short, the distributions follow the generation way in [1] - they are either Gaussian mixtures or uniform distribution. We sample 8000 data points randomly for each component distribution, and train 1 diffusion model for each distribution. We will include more parameters of the distributions in Appendix D.
>
> > Unclear noise sharing performance loss in negation
>
> Our hypothesis is that sharing the same noise introduces some bias in the estimation, and it makes CompLift over-reject samples as a conservative way. With more trials, the bias of the estimation is amplified, thus makes more samples over rejected. After taking a deeper look in Figure 2, we also observe similar  sharing-noise regression for Product and Mixture, though the regression is very slight for those 2 algebras. We will modify the explanation in the paper to make this hypothesis more clear.
>
> > Cached CompLift description unclear
>
> We provide more details about the algorithm in Appendix C. Algorithm 5 and 6 are in pseudo-code styles and might be easier for the reviewer to parse. We will add a sentence in Section 4.3 to refer readers to Appendix C for more context. Please let us know if there remains such an issue, we will keep making the paper easier to read.
>
> > EBM-related abbreviations
>
> Thanks. We'll add explanations in Section 6.1 for all abbreviations (EBM, ULA, U-HMC, MALA, HMC).
>
> > Typos
>
> Thanks. We'll remove the self-reference and change $z$ to $z_t$ in paragraph L317.
>
> [1] https://arxiv.org/abs/2302.11552

---

> > ### Comment · Reviewer_kdTw · 2025-04-03
> >
> > I appreciate the rebuttal from the authors that addresses all my concerns from my review. I do not have any follow-up questions.

---

### Official Review · Reviewer_kyUu · 2025-03-13

**Overall Recommendation:** 3

**Summary:**

This work proposes CompLift, a resampling criterion based on the concept of lift scores used to improve the compositional generation capabilities of pretrained diffusion models. CompLift approximates the lift scores with the diffusion modules noise estimation, without requiring any external reward modules to measure its alignment to the given condition. The authors additionally propose a caching technique for CompLift and achieves computationally effecient pipeline. Through evaluations on both simple synthetic generation and more complex text-to-image generation, the paper shows that CompLift leads to accurate compositional generation without additional training.

**Claims And Evidence:**

The overall writing of the paper is well-structured, with a clear problem definition and a simple but effective solution. The idea of adopting the concept of lift scores for improving diffusion model's compositional generation is interesting.

However, the paper lacks reference and discussions on an important related work, as detailed below:

CompLift seems to resemble a closely related work CAS [1], in which the authors define a novel "condition alignment score" CAS as $\log p (x_0 | c) - \log p (x_0)$ (Fig. 3 of the paper). The main argument of CAS is that this term can effectively measure the alignment between the generated output $x_0$ and the given condition $c$, and therefore it can be used as a alignment metric without the need of external modules. This claim is similar to the main contribution of this work.

In this regard, the proposed formulation of lift scores in Eq. (2)-(4) of this work seems to be quite similar to CAS, and the authors will need to provide a discussion on the difference between the two approaches in order to claim the novelty of CompLift.

[1] CAS: A Probability-Based Approach for Universal Condition Alignment Score, Hong et al., ICLR 2024

**Essential References Not Discussed:**

A critical related work CAS [1] is missing, as mentioned in the "Claims" section.

[1] CAS: A Probability-Based Approach for Universal Condition Alignment Score, Hong et al., ICLR 2024

**Experimental Designs Or Analyses:**

1. Comparisons on the running time in Fig. 5 clearly shows the advantage of CompLift over the MCMC-based approaches in terms of efficiency.

2. The idea of "counting the activated pixels" in Section 5.1 seems quite confusing. I'm curious whether this design choice can consider a variety of objects. For instance, some objects are likely to take up a large area of the image, while other objects might be generated in smaller sizes. If the same threshold for number of pixels is applied for checking the existence, can it handle both cases? How did you set the threshold $\tau$?

**Methods And Evaluation Criteria:**

1. For the request for the justification on the novelty of CompLift, please refer to the above section.

2. Evaluations on the effect of CompLift on text-to-image generation is not very convincing as it does not include comparisons with the baselines. While the authors evaluate using the benchmark from Attend-and-Excite [1], I couldn't find the comparisons with the Attend-and-Excite itself. Since Attend-and-Excite (or its follow-ups) also does not require external modules for measuring the alignment with the given condition, I believe it should be a valid baseline for comparison. Otherwise, as stated in the introduction, it would be nice if the authors show that CompLift can indeed to applied together with such methods, yielding additional improvements.

[1] Attend-and-Excite: Attention-Based Semantic Guidance for Text-to-Image Diffusion Models, Chefer et al., SIGGRAPH 2023

**Other Comments Or Suggestions:**

typo: page 8 line 408: ImageResizer -> ImageReward

**Other Strengths And Weaknesses:**

I agree with the fundamental goal of the paper that "diffusion models should be able to assess its own alignment to the given condition". And the method is quite simple and intuitive. However, for now it is hard to give a positive score as the paper fails to address a critical previous work that has proposed a similar claim and solution.

**Questions For Authors:**

Crucial questions are included in the "Claims" section.

**Relation To Broader Scientific Literature:**

As also mentioned in the paper, the idea of using the diffusion model itself as a source of a conditional reward function could be useful for inference-time scaling methods for diffusion models aiming for better condition alignment.

**Theoretical Claims:**

This paper doesn't provide theoretical claim or proof, and instead focuses on the empirical evidence on numerous types of data.

---

> ### Author Rebuttal · Authors · 2025-03-30
>
> Thank you for your thoughtful and constructive feedback. We address your concerns as follows.
>
> > Relationship to CAS [1]
>
> Thank you for pointing out this important related work, which we previously overlooked. We will add reference to this valuable work, and incorporate a discussion of CAS in the "Related Work" section of the revised version. A summary of the relationship is as follows:
>
> Our work can be seen as an extension of CAS, which investigates the potential of using CAS as a compositional criterion to decompose the alignment requirements of a complex prompt into multiple acceptance criteria. To approximate CAS, we employ ELBO estimation to reduce computational cost, as an alternative to the Skilling-Hutchinson estimator used in the original CAS paper. It would be interesting to explore how Skilling-Hutchinson-based estimation performs in a compositional setting. It may yield higher accuracy at the expense of greater computational overhead, which we plan to investigate in future work.
>
> > Comparison with Attent-and-Excite [2] on text-to-image generation
>
> Thank you for the great question. We conducted a new experiment using Attent-and-Excite [2], focusing on the additional improvement achieved by incorporating CompLift. We observed consistent performance gains with both SD 1.4 and SD 2.1. Note that SD XL is not included due to the lack of support in the original Attent-and-Excite code.
>
> | Method | Animals |  | Object & Animal |  | Objects |  |
> | :---: | :---: | :---: | :---: | :---: | :---: | :---: |
> |  | CLIP ↑ | IR ↑ | CLIP ↑ | IR ↑ | CLIP ↑ | IR ↑ |
> | A&E (SD 1.4) | 0.330 | 0.831 | 0.357 | 1.339 | 0.357 | 0.815 |
> | A&E (SD 1.4) + *Cached CompLift* | **0.338** | 1.156 | **0.361** | **1.469** | **0.362** | 0.934 |
> | A&E (SD 1.4) + *CompLift* | 0.337 | **1.160** | **0.361** | 1.458 | 0.361 | **0.990** |
> ||||||||
> | A&E (SD 2.1) | 0.342 | 1.225 | 0.360 | 1.471 | 0.366 | 1.219 |
> | A&E (SD 2.1) + *Cached CompLift* | 0.344 | 1.298 | 0.364 | 1.488 | **0.371** | 1.245 |
> | A&E (SD 2.1) + *CompLift* | **0.346** | **1.337** | **0.365** | **1.516** | 0.370 | **1.246** |
>
> > The idea of "counting the activated pixels" in Section 5.1 seems quite confusing... How did you set the threshold $\tau$?
>
> Thank you for raising this thoughtful concern. We agree that performance could be further improved by making $\tau$ an object-specific hyperparameter. For simplicity, we currently set $\tau$ as a uniform threshold.
>
> We chose $\tau = 250$ as the median value among the number of activated pixels across all images. We also experimented with the 25th and 75th percentiles, and found the median performed best in practice. A lower $\tau$ leads to less accurate rejection due to ELBO variance, while a higher $\tau$ increases the rejection rate.
>
> We will include more details on the derivation of $\tau$ in the Appendix. Intuitively, $\tau = 250$ corresponds to ~1.5% of the total number of the latent pixels in SDXL (128x128 latent space) and ~6.1% in SD 1.4/2.1 (64x64). We find this small threshold sufficient, since our focus is on identifying "missing object" issues. If an object is missing, it tends to result in almost no activated pixels. Additional discussion will be added to the Appendix.
>
> > Does the same trend hold for the recent work "Reduce, Reuse, Recycle" [3] based on MCMC?
>
> Thank you for this suggestion. We also observed a clear overall advantage of our method over MCMC-based methods like "Reduce, Reuse, Recycle" [3]. While certain MCMC variants such as U-HMC and MALA perform comparably in specific scenarios (e.g., the first test case in Product and second in Mixture), they often generate samples outside the target distribution in other settings, unlike our method.
>
> We will update Appendix D to include visualizations comparing results from the MCMC-based methods.
>
> > Typo: page 8 line 408: ImageResizer -> ImageReward
>
> Thank you for catching this typo. We will correct it in the revised version.
>
> [1] https://openreview.net/forum?id=E78OaH2s3f
>
> [2] https://arxiv.org/abs/2301.13826
>
> [3] https://arxiv.org/abs/2302.11552

---

> > ### Comment · Reviewer_kyUu · 2025-04-05
> >
> > I appreciate the authors' rebuttal and their efforts in answering all the raised questions. However, I still have concerns regarding the core technical contribution of LiftScore over CAS, outlined below:
> >
> > While the authors distinguish between conditional generation (in CAS) and compositional generation (in LiftScore), it is unclear whether these tasks are fundamentally independent. The tasks in LiftScore (e.g. text-to-image generation, position task) could be framed as conditional generation tasks, which means that they can also be addressed by CAS.
> >
> > Could the authors clarify why LiftScore could have an advantage over CAS specifically in the compositional setting?
> >
> > If the key difference between the two methods is the choice of the approximation, I am concerned whether this choice can be justified for the specific tasks.

---

> > > ### Author Response · Authors · 2025-04-05
> > >
> > > Thank you for your acknowledgement of our rebuttal effort. We address your question as follows:
> > >
> > > > advantage of compositional criteria
> > >
> > > We acknowledge that the compositional acceptance / rejection task can also be framed using 1 single criterion that works directly on the whole prompt, as addressed by CAS. To test how CAS-like variant performs for prompts containing multiple objects, we have conducted a new ablation study.
> > >
> > > Here, CAS variant means that we use the single criteria $\log p(z| c_{\text{compose}})-\log p(z| \varnothing)$ as the latent lift score, which replaces the composed criteria from multiple individual lift scores in CompLift. Note that this is a controlled experiment to check the advantage of compositional criteria, thus, we keep the same estimation method using ELBO.
> > >
> > > We provide the following table as the result. We observe only modest improvement when using the CAS variant. We hypothesize that CAS variant might face a similar problem as the original Diffusion Model for multi-object prompts - the attention to the missing object is relatively weak in the attention layers. Similar discussions can be found in previous works such as Attend-and-Excite [1], where diffusion model $\epsilon_\theta(x, c_\text{compose})$ sometimes has weak alignment with some condition $c_i$ in $c_\text{compose}$.
> > >
> > > We will add the new table and the related discussion to the Appendix.
> > >
> > > | Method | Animals |  | Object&Animal |  | Objects |  |
> > > | --- | :---: | :---: | :---: | :---: | :---: | :---: |
> > > |  | CLIP ↑ | IR ↑ | CLIP ↑ | IR ↑ | CLIP ↑ | IR ↑ |
> > > | SD 1.4 | 0.310 | -0.191 | 0.343 | 0.432 | 0.333 | -0.684 |
> > > | SD 1.4 + *CAS Variant* | 0.312 | -0.153 | 0.348 | 0.708 | 0.337 | -0.373 |
> > > | SD 1.4 + *CompLift* | **0.322** | **0.292** | **0.358** | **1.094** | **0.347** | **-0.050** |
> > > ||||||||
> > > | SD 2.1 | 0.330 | 0.532 | 0.354 | 0.924 | 0.342 | -0.112 |
> > > | SD 2.1 + *CAS Variant* | 0.333 | 0.626 | 0.355 | 1.080 | 0.347 | 0.144 |
> > > | SD 2.1 + *CompLift* | **0.340** | **0.975** | **0.362** | **1.283** | **0.355** | **0.489** |
> > > ||||||||
> > > | SD XL | 0.338 | 1.025 | 0.363 | 1.621 | 0.359 | 0.662 |
> > > | SD XL + *CAS Variant* | 0.338 | 1.064 | 0.363 | 1.628 | 0.362 | 0.702 |
> > > | SD XL + *CompLift* | **0.342** | **1.216** | **0.364** | **1.706** | **0.367** | **0.890** |
> > >
> > > > side note: why compositional in general?
> > >
> > > One cause of the missing object issue might be the training-inference mismatch: similar combinations of objects in $c_\text{compose}$ are rare in the training set. As more objects of interest are involved, we can observe that this problem gets more significant as the whole composed condition grows more complex (e.g., the CLEVR experiment). Similarly, sampling/criteria based solely on $\epsilon_\theta(x, c_\text{compose})$ might not be as reliable as approaches that incorporate information from individual $\epsilon_\theta(x, c_i)$, as the table shown above.
> > >
> > > Compositional generation is one approach to generalize to more complex prompts. For example, the CLEVR model is trained with only 1 object position in the prompt. With Composable Diffusion + CompLift, we can extend it to the combination with 5 object positions with high accuracy, while such combinations are rare to see in training.
> > >
> > > > key core contribution of our work
> > >
> > > We would like to emphasize that our key contribution is to provide a systematic exploration and application of LiftScore / CAS, specifically to compositional generation challenges. Our contribution is not the invention of LiftScore, since it is already an existing concept in data mining, and theoretically equivalent to the CAS concept.
> > >
> > > Our work can indeed be viewed as a complementary extension of LiftScore / CAS into the compositional generation domain - much like how science builds upon previous discoveries, we too stand on the shoulders of giants.
> > >
> > > The CAS paper provided valuable insights on condition alignment with a single condition, which we acknowledge. Our contribution extends this foundation by:
> > >
> > > 1. Developing the mathematical framework to apply these scores to multiple condition compositions, including algebras like Product, Mixture, and Summation.
> > > 2. Introducing novel engineering solutions (like caching and variance reduction) that make compositional evaluation practical.
> > > 3. Systematic evaluation on 2D, CLEVR, and text-to-image datasets.
> > >
> > > We hope this clarification addresses your concerns about the relationship between our work and CAS. Our intention is to contribute meaningful extensions to this line of research by adapting and enhancing these techniques specifically for compositional generation tasks.
> > >
> > > Thank you again for your insightful feedback, which has helped us better articulate the positioning of our work. If our responses have addressed your concerns adequately, we would be sincerely grateful if you would consider raising your score accordingly as a recognition of our work and this rebuttal effort. Thank you once again for your time and support.
> > >
> > > [1] https://arxiv.org/abs/2301.13826

---

### Official Review · Reviewer_25c6 · 2025-03-13

**Overall Recommendation:** 3

**Summary:**

This paper proposes a training-free post-processing approach, CompLift, to select images with specified concepts from diffusion model-generated image candidates. The main idea is to use the lift score, which is equivalent to point-wise mutual information, to evaluate if conditioning $c$ reduces uncertainty of variable $\textbf{x}$. As a post-processing approach, the performance of CompLift hinges on the generative model (i.e., composable diffusion model) it is based on. If composable diffusion model cannot generate accurate images at all, then CompLift cannot make any improvement. Experimental results show improved generation accuracy by using the proposed approach.

### Most of my concerns are addressed. I maintain the rating.

**Claims And Evidence:**

The paper claims that "as a novel resampling criterion using lift scores for compositional generation, requiring minimal computational overhead". This assertion seems somewhat overstated, and the term "minimal" is ambiguous without a clear criterion. While in some cases, the cached strategy results in no additional computational overhead, this is not universally true.  For text-to-image generation, when replacing $\epsilon$ with $\epsilon_\theta(z, c_{\text{ccompose}})$, additional computational overhead (n + 2) · T forward passes is involved.

**Essential References Not Discussed:**

N/A

**Experimental Designs Or Analyses:**

The paper states in Fig. 5 that the overhead introduced by the cached CompLift is negligible for the Composable Diffusion baseline (Liu et al., 2022). However, this experiment only shows running time without giving accuracy evaluation. It is not clear if it is trading accuracy performance with running time. It might be helpful to show both in a single figure.

**Methods And Evaluation Criteria:**

The proposed method make sense for the application.

**Other Comments Or Suggestions:**

N/A

**Other Strengths And Weaknesses:**

**Strength**

1. The proposed approach is training-free and requires little or no additional computational resources by cache design.

2. Extensive experiments are conducted and show improvements over baselines.

3. The writing is smooth and easy to follow.

**Weakness**

1. Involving no extra training or guidance at inference time not only can be an advantage but also can be a limitation of the proposed model. This means that the proposed model cannot correct the generated images but can only select some from them. As a result, the performance of CompLift largely hinges on the generative model (like Composable Diffusion) it builds upon, because if Composable Diffusion does not generate accurate images, the CompLift cannot select accurate images from the generated images.

2. Though CompLift shows significant improvement over Composable Stable Diffusion on synthetic datasets, the improvement on real-world text-to-image generative model is trivial as shown in Table 3. Again, this hinges on the performance of Composable Stable Diffusion that has limited ability to generate accurate multi-object images.

3. Though called CompLift, the proposed approach itself is not compositional because it evaluate individual concepts separately. It is not evaluating a joint appearance of all concepts. For text-to-Image generation, only AND operation is considered by evaluating individual concepts.

4. For text-to-image generation, when replacing $\epsilon$ with $\epsilon_\theta(z, c_{\text{ccompose}})$, additional computational
overhead (n + 2) · T forward passes is involved. This contradicts the minimal computational overhead requirement claim, and should be discussed to avoid overclaim.

**Questions For Authors:**

It is somewhat unclear about fair comparison with Composable Diffusion Model. Consider that Composable Diffusion Model generates 5 images, and one of them is accurate. CompLift selects the accurate one with lift score. Then how to determine if Composable Diffusion Model or CompLift is more accurate?

**Relation To Broader Scientific Literature:**

The proposed approach uses lift score as a criterion to evaluate if a concept appears in an image. The estimated lift score in equation (4) is actually equivalent to the point-wise mutual information discussed in equation (5) in [1]. The proposed approach can be seen as an application of point-wise mutual information in [1].

[1] Kong, X., Liu, O., Li, H., Yogatama, D., and Steeg, G. V. Interpretable diffusion via information decomposition. arXiv preprint arXiv:2310.07972, 2023.

**Theoretical Claims:**

No theoretical claims are provided in the paper.

---

> ### Author Rebuttal · Authors · 2025-03-30
>
> Thank you for your valuable feedback and questions. We address your concerns below:
>
> > On CompLift's dependence on underlying generative model
>
> We agree and will add this theoretical limitation to our Conclusion. While theoretically CompLift cannot improve if the base method produces no accurate images, in practice even weak generators often improve with ≤5 candidate images.
>
> > On "minimal computational overhead" claim
>
> We'll remove the ambiguous term "minimal" and claim only "requiring no additional training." For text-to-image generation, the (n + 2) · T additional forward passes can be parallelized to reduce latency. Currently, generation takes ~15s and lift score calculation ~30s on a 4090 GPU, with GPU memory as the bottleneck. Ideally, we can further parallelize to the latency of O(1) forward pass given enough GPU memory.
>
> > On cached CompLift's accuracy-speed tradeoff
>
> Every column in Fig. 5 has a corresponding accuracy row in Table 2 (Cached CompLift has T=50 by default). We'll add a footnote clarifying this. In practice, we perceive small accuracy regression on Mixture and Negation task when switching from vanilla CompLift to Cached CompLift. However, the accuracy remains significantly higher than other baselines. The tradeoff exists, but seems to be mild and acceptable given the substantial speed improvement.
>
> > On lift score equivalence to point-wise mutual information [1]
>
> We'll update the Related Work section to reflect this equivalence. Our paper applies point-wise mutual information (PMI) as an acceptance/rejection criterion, focusing on missing-object cases and composing PMI for multiple objects.
>
> > On real-world text-to-image improvement
>
> The seemingly trivial CLIP improvement is due to the low magnitude of CLIP scores. Here, we provide another perspective to interpret the numbers. We compare the CompLift selector to the perfect best-of-n selector, which has direct access to the metric function. The percentage gain is calculated as (CompLift metric - baseline metric) / (perfect selector metric - baseline metric). On average, the gains are ~40% for vanilla CompLift and ~30% for cached CompLift. We will add more explanation to the Appendix.
>
> | Method | Animals |  | Object&Animal |  | Objects |  |
> |--------|---------|---------|---------|---------|---------|---------|
> |        | CLIP gain% ↑ | IR gain% ↑ | CLIP gain% ↑ | IR gain% ↑ | CLIP gain% ↑ | IR gain% ↑ |
> | SD 1.4 + *Cached CompLift* | 31.58 | 30.58 | 42.62 | 62.74 | 37.04 | 54.99 |
> | SD 1.4 + *CompLift* | **42.11** | **46.40** | **49.18** | **74.32** | **47.14** | **63.05** |
> ||||||||
> | SD 2.1 + *Cached CompLift* | 35.71 | 44.13 | 28.34 | 55.07 | 37.97 | 57.82 |
> | SD 2.1 + *CompLift* | **39.68** | 56.18 | **32.39** | **60.28** | **41.14** | **74.73** |
> ||||||||
> | SD XL + *Cached CompLift* | 16.95 | **55.88** | **26.13** | 46.15 | 24.49 | **46.06** |
> | SD XL + *CompLift* | **22.60** | 48.74 | **26.13** | **59.44** | **32.65** | 44.88 |
>
> > On CompLift's compositionality
>
> Our approach is to (1) evaluate individual concepts separately, and (2) compose an acceptance/rejection criteria from these multiple individual criteria, as Algorithm 3. Thus, the CompLift criteria seems compositional from our perspective. We wish to mention Composable Diffusion [2] as an example to elucidate such a perspective. Essentially, the approach uses (1) individual score on each concept, and (2) compose the score from these multiple individual scores. Such a factorize-and-compose way helps reduce the hardness to comply with complex prompt. As shown in our experiments, it improves performance on complex multi-object generation by evaluating individual conceptual alignment before making a composed decision.
>
> > On limited algebraic operations for text-to-image
>
> We acknowledge this limitation. While we tested all algebras in the 2D dataset, we found no existing mature benchmark for OR/NOT algebras in text-to-image generation. We'll note this in our Conclusion for future work.
>
> > On fair comparison with Composable Diffusion
>
> We recognize the challenge in comparing these approaches. Composable Diffusion generates candidates with no internal selection mechanism, while CompLift is a post-hoc filter. They serve different purposes and are not mutually exclusive—CompLift can enhance generation models by leveraging semantic alignment for filtering results. Our main goal in the experiments is to show such an enhancement, instead of a direct replacement, of other baselines such as Composable Diffusion.
>
> [1] https://arxiv.org/abs/2310.07972
>
> [2] https://arxiv.org/abs/2302.11552

---

### Decision · Program_Chairs · 2025-05-01

**Decision:**

Accept (poster)

**Comment:**

The reviewers find the paper well written, and the evaluation demonstrates that the proposed method consistently outperforms the baseline, particularly on the synthetic dataset. However, as noted by Reviewer kyUu, the literature review is incomplete, with a highly relevant work omitted which impacts the novelty and contribution of the paper. Additionally, the reviewers find the evaluation on real-world data less convincing. These concerns were addressed in the rebuttal, leading to improved overall recommendations. Area Chair agrees with the reviewers and considers the paper above the acceptance threshold, provided the authors appropriately incorporate the content and clarifications presented in the rebuttal into the final revision.